

# Mediterranean seagrasses as carbon sinks: Methodological and regional differences

Anna Escolano-Moltó[1], Susana Flecha[1,2], Raquel Vaquer-Sunyer[3], Marlene Wesselmann[1], Núria Marbà[1], Iris E. Hendriks[1]

[1]Instituto Mediterráneo de Estudios Avanzados (IMEDEA-CSIC-UIB), C/Miquel Marqués 21, 07109 Esporles, Illes Balears, (Spain).
[2]Instituto de Ciencias Marinas de Andalucía (ICMAN-CSIC), Campus Universitario Río San Pedro s/n, 11519 Puerto Real, Cádiz (Spain).
[3]Marilles Foundation, Bisbe Perelló 1, 7, 07002, Palma de Mallorca, Illes Balears, (Spain).

*Correspondence to* Anna Escolano-Moltó (Ana.EscolanoMolto@imbrsea.eu)

**Abstract.** The increasing rates of $CO_2$ due to anthropogenic activities are causing important potential climate threats for the Mediterranean Sea: ocean acidification and warming. In this region, two seagrass species, *Posidonia oceanica* and *Cymodocea nodosa* can play a crucial role in climate change mitigation. Through their metabolic activity, they can act as carbon sinks; buffer lowering pH values during the day and store carbon in the sediment underneath their meadows. In this study we analyse the metabolism synthesized from published data on seagrass community metabolism and from own results to evaluate trends through time of these two species comparing two methodologies: benthic chambers and multiparametric sensors. Furthermore, we analysed seasonal trends of both seagrass species´ metabolic rates and their variation between the Eastern and Western Mediterranean basins, with no significant results despite the clear visual trends. Our analysis revealed that there are significant differences between methodologies, with multiparametric sensors estimating higher rates, but unable to differentiate between habitats and useful to assess seagrass metabolism at a community level whereas benthic chambers are capable to evaluate rates at a seagrass species level. We found significant differences between the two Mediterranean regions for both methodologies, with highest rates of Net Community Production found in the Easter basin. At a species level, we found that Posidonia was more productive compared to Cymodocea. Furthermore, 86.7% of the metabolic values reflected that the meadows were acting as carbon sinks in the Western basin.

## 1 Introduction

Organic carbon buried in marine vegetation, like mangroves, saltmarshes and seagrass sediments is known as "blue carbon". Despite the fact that seagrass meadows cover only a 0.1% of the ocean surface, they are responsible of a 20% of the global carbon sequestration in marine sediments (Duarte et al., 2004; Kennedy et al., 2010), acting as important key players in "blue carbon".(Duarte et al., 2004; Kennedy et al., 2010; Mcleod et al., 2011; Greiner et al., 2013). Carbon burial is the result of intense metabolic activity, together with excess of production, high trapping capacity of allochthonous matter in seagrass meadows and an increased carbon preservation in sediments (Cebrian, 1999). Particles from the water column containing carbon and other elements such as nutrients like nitrogen and phosphorus are buried beneath the seagrass meadows. This is



due to the enhanced deposition as canopies capture suspended organic matter in the water column, accumulating it as organic matter in the sediment (Romero et al., 1994; Pergent et al., 1997; Mateo et al., 2006; Hendriks et al., 2008; Kennedy et al., 2010), together with *in situ* production due to their primary production (Greiner et al., 2013). There are species specific

differences in carbon accumulating capacities, for instance the organic-rich materials accumulated beneath a *P. oceanica* canopy can be up to 6000 years old and reach a thickness of up to 13 metres and for this species a huge carbon storage capacity has been estimated, ranging from 40 to 770 $kg\,C_{org}\,m^{-2}$ (Mateo et al., 1997; Lo Iacono et al., 2008; Serrano et al., 2016). For *Cymodocea nodosa,* the annual carbon budget has been estimated on $4.4\,g\,C\,m^{-2}\,y^{-1}$ whereas the Posidonia oceanica has been estimated to be $66.4\,g\,C\,m^{-2}\,y^{-1}$ in the same location (Cebrián et al., 1997). Seagrass communities tend to be autotrophic,

nevertheless primary production can vary depending on temporal scales: daily, seasonally (Bay, 1984; Alcoverro et al., 1995; Gobert et al., 2006; Hendriks et al., 2014) and year to year (Champenois et al., 2012, 2019). Furthermore, the variation of primary production and carbon storage depends on many other variables such seagrass habitats (Lavery et al., 2013; Alongi et al., 2016), structural complexity (Trevathan-Tackett et al., 2015), nutrients dynamics (Armitage et al., 2016), hydrodynamics (Samper-Villarreal et al., 2016), water depth (Serrano et al., 2014) or size of the meadow (Ricart et al., 2017). Consistent

estimates of seagrass meadows productivity are crucial to estimate the contribution to the global carbon sink capacity of the biosphere and to approximate the economic and ecological consequences of their decline worldwide (Orth et al., 2006; Waycott et al., 2009). Despite its crucial importance, little is known about how the increasing rates of atmospheric $CO_2$ are going to affect those invaluable ecosystems. Predicted open ocean conditions may not reflect the future in coastal zones (Hendriks et al., 2010; Hofmann et al., 2011; Kelly et al., 2013; Lacoue-Labarthe et al., 2016). pH changes in coastal ecosystems are

complex and englobe different drivers with complex biogeochemical dynamics ruled by interactions between processes on land, open ocean and atmosphere (Aufdenkampe et al., 2011), which are often affected by human processes. Dynamics in coastal regions are frequently influenced by benthic ecosystems that have the capacity to temper physical and chemical conditions of the environment (Gutiérrez et al., 2011). By absorbing $CO_2$ and producing $O_2$, these highly productive systems cause variations in pH and dissolved oxygen concentrations in the adjacent water column that follows daily and seasonal

patterns modulated through metabolic activity (Duarte et al., 2013; Hendriks et al., 2014).. This pH variation is complex and depends on the balance between the absorption of atmospheric anthropogenic $CO_2$, inputs of organic matter, watershed export of alkalinity and the variations in the balance between calcification rates, respiration and primary production (Duarte et al., 2013). Primary production is hence an important component in the assessment of pH variation in coastal ecosystems. Through their photosynthetic activity, pH modification of the adjacent water mass by seagrasses attenuates and buffers ocean

acidification, providing protection for calcifying organisms and oxygenating the water column through $O_2$ production (Hendriks et al., 2014). Through their metabolic activity, seagrass meadows can sequester and store a considerable amount of carbon in the sediments (Mcleod et al., 2011; Fourqurean et al., 2012). Assessing their metabolism is crucial to understand their contribution as blue carbon sinks and water oxygenators. Seagrass metabolism can be assessed through the concentration of dissolved oxygen in water. Water temperature has an effect in the air-water exchange by modifying oxygen solubility and

has an impact on oxygen dynamics by affecting the ecosystem metabolism (Brown et al., 2004). In addition, anthropogenic



pressures such as eutrophication (Paerl, 2006) or climate change are affecting oxygen cycling (Keeling et al., 2002; Conley et al., 2009; Keeling et al., 2010). Indeed, dissolved oxygen is one of the environmental parameters that has changed more drastically in a short period of time (Diaz et al., 1995; Diaz, 2001) with potential catastrophic consequences for marine life (Vaquer-Sunyer et al., 2008). In coastal ecosystems, increased nutrient inputs contribute to higher organic production and

oxygen demand with a consequently greater hypoxia likelihood (Karim et al., 2003; Zhang et al., 2010). Measuring dissolved oxygen concentrations allows to estimate seagrass metabolic parameters such as Gross Primary Production (GPP). At a global scale, GPP of seagrass meadows can vary from 296 to 591 Tg C yr$^{-1}$ with a net organic carbon burial ranging between 25 and 50 Tg C yr$^{-1}$ (Duarte et al., 2010). This different fate of autochthonous organic carbon will depend on the metabolic community status, which is defined by the Net Community Production (NCP), the difference between GPP and Community Respiration

(CR). When a community exports organic carbon to adjacent systems, it is considered as net autotrophic (NCP > 0), on the other hand if the community gets external organic carbon inputs to sustain its own metabolism, it is considered as net heterotrophic (NCP < 0) (Champenois et al., 2012). If the seagrass system is net autotrophic, gross primary production exceeds respiration (Duarte et al., 2010), in which photosynthetic activity supports high net community production and efficient sequestration of carbon biomass (Gullström et al., 2018).

In this work, we studied the metabolism of the seagrass species *Cymodocea nodosa* and *Posidonia oceanica*, which are located in the Mediterranean Sea, a region that has been defined as a "hotspot for climate change" (Giorgi, 2006) with the highest rates of warming, two- to four-fold higher than in other regions (Vargas-Yáñez et al., 2008; Vaquer-Sunyer et al., 2010; Burrows et al., 2011). Jordà et al. (2012) projected a rise in the seasonal average temperature of 3.4ºC in summer and 2.2ºC in winter by the end of this century for the Mediterranean Sea Western Basin. An increase of 2 to 3ºC in the Sea Surface Temperature (SST)

between 1982-2003 (Belkin, 2009; Richon et al., 2019) has been detected in the Levantine (Eastern) Basin. Observations from 1985-2006 from Nykjaer (2009) indicate that in the last two decades SST has been increasing at an average (±SD) rate of 0.050 ±009ºC yr$^{-1}$ in the Eastern basin and 0.03±0.008ºC yr$^{-1}$ for the Western Basin. Indeed, Amitai et al. (2020) have demonstrated that the SST anomaly in the Eastern Mediterranean surface temperature is increasing considerably faster than in the Western Mediterranean. Furthermore, extreme thermal events are expected to be more intense and frequent in the Mediterranean region

(IPCC, 2013).

In terms of distribution, the total surface area occupied by *P.oceanica* meadows is estimated as ranging from 1 to 2% of the total surface area of the Mediterranean Sea (Béthoux et al., 1986; Pasqualini et al., 1998) although this number is uncertain (Bonacorsi et al., 2013). The current distribution of *P. oceanica* meadows in the Mediterranean Sea is estimated as 510.710 ha and 713.992 ha in the Western and Eastern basins, respectively (Telesca et al., 2015). In their report, historical and present

data was analysed, highlighting the remarkable differences between the existing data in the Western and Eastern basins: there was and there is much more data available in the Western basin compared with the Eastern part where absence of data is common. As with other seagrasses around the world (Waycott et al., 2009), the extent of Mediterranean meadows is decreasing considerably (Boudouresque et al., 2009; Marba et al., 2014; Telesca et al., 2015). In the last 50 years, the estimated regression of meadows in the Mediterranean Sea reached 34% of the total distribution (Telesca et al., 2015). There is evidence that climate



change can impact *P. oceanica* meadows negatively, as higher temperatures stress the species physiologically (Marbà et al., 2010), with shoot mortality increasing during heat waves (temperatures greater than 28°C) at the end of summer (Diaz-Almela et al., 2007). These water temperatures are likely to increase with global warming (Jordà et al., 2012). In the case of *C. nodosa* meadows, their thermal tolerance is higher and they are supposed to cope better with increasing temperatures (Egea et al., 2018). Nevertheless, Olsen et al. (2012) demonstrated in their study that high temperatures during heat waves over coming

decades will also have a significant negative impact on Mediterranean *C. nodosa* populations. A future 4ºC-change in the annual mean temperature as a consequence of heat waves, will probably exceed the limit beyond which *C. nodosa* loses can be expected in the Mediterranean Sea (Olsen et al., 2012). Moreover, all Mediterranean water bodies are contaminated by anthropogenic carbon but the Western basin is more contaminated than the Eastern basin and all waters have been acidified by values ranging from -0.14 to -0.005 pH units since the beginning of the industrial era to 2001, clearly higher than elsewhere

in the open ocean (Touratier et al., 2011). Indeed, ocean acidification is a climate change indicator that has been characterized as one of the most important climate warnings for the Mediterranean Sea, together with temperature and UV radiation (Micheli et al., 2013). In addition to climate change, the Mediterranean Sea is likely to be more impacted by disturbances than other seas (Giorgi et al., 2008; Richon et al., 2019). For instance by natural and anthropogenic (non-related to climate change) impacts, such as overfishing, increasing pollution levels and the introduction of alien species (Lejeusne et al., 2010). In

addition, these ecosystems can be extremely affected by a inputs derived from human activity as wastewater outfalls, riverine, farmland, runoffs and fish farming (Apostolaki et al., 2007; IPCC, 2013; Powley et al., 2016). It is crucial then to evaluate the health status of key ecosystems such as seagrasses. Estimating seagrass metabolism can be one approach to assess the health status of these communities.

    Seagrass metabolism has classically been measured by the use of closed benthic chambers. However, the spatial heterogeneity

of these ecosystems (Gazeau et al., 2005) and their high temporal variability cannot be easily estimated with this approach (Karl et al., 2003). For this purpose, sensors methodology can be more suitable as dissolved oxygen concentrations can be estimated through longer periods of time. In addition, seagrass metabolism has been estimated recently with the non-invasive aquatic eddy covariance technique, nevertheless there is only one study including *P.oceanica* (Koopmans et al., 2020) in the Mediterranean Sea. GPP values obtained with the use of benthic chambers could provide underestimates as a result of

photorespiration, while the use of multiparametric probes measuring oxygen in the canopy probably provides more realistic GPP values (Champenois et al., 2012). The use of multiparametric probes to measure $O_2$ also provides the opportunity to measure metabolic rates in a relatively easy way and during large periods. In river and lake ecosystems, the measurement of metabolism by oxygen probes and loggers is a method generally used (Cole et al., 2000; Coloso et al., 2008), while it is not widely spread in coastal waters (Odum et al., 1958; Odum et al., 1962; Ziegler et al., 1998; Vaquer-Sunyer et al., 2012) due to

higher lateral transport rates of water in these systems.

    Therefore, the aim of this study is to evaluate the potential of the two Mediterranean Sea seagrass species *Posidonia oceanica* and *Cymodocea nodosa* as carbon sinks through their metabolic activity, comparing two methodologies (benthic chambers and multiparametric probes). In order to evaluate the spatial and temporal metabolic activity between Mediterranean regions the



existing literature including the two species in the Mediterranean Sea has been considered. The analysis performed in this work
represents a relevant contribution of the role of these valuable ecosystems in the context of global change mitigation.

## 2 Methods

### 2.1. Data compilation for multiparametric probes

In this study, part of the sensors data for the metabolic parameters was directly extracted from literature and the other part was
obtained from published datasheets, processed and analysed to obtain the metabolic parameters.

### 2.1.1 Site description

We estimated metabolism from oxygen data of multiparametric sensors deployed in the Western and Eastern Mediterranean
basin. In total we processed data from eight sites in Mallorca (Spain), two sites in Crete (Greece) and one in Cyprus (Republic
of Cyprus). Sampling campaigns were carried out during different periods starting from 2011 to 2019 (for details see Table 1).
All study sites were located in shallow sites, ranging from 2.9 metres depth (Punta Negra, Mallorca) to 15.7 metres depth (Cap
Enderrocat, Mallorca). Multiparametric probes were measuring in either *Posidonia oceanica* and/or *Cymodocea nodosa*
meadows (see Table 1).

The sampling site in Cyprus was located in Limassol, East Akrotiri bay, considered an impacted area affected by high
anthropogenic pressures related to tourism and the construction of extensive coastal infrastructures. In Crete, Marathi and
Kalami are considered as a single sampling site due to the proximity and similitude of the environmental factors of both
sampling sites. This sampling station, located in Western Crete close (< 10 km) to the Port of Souda, is impacted by notably
sewage discharge, agriculture and industrial/chemical pollution; according to Simboura et al. (2016) this station is considered
to have a moderate pressure index. Maridati, the second station located in Crete is situated on the East side of the Island, in a
pristine bay with no human coastal activity but affected by ensuing discharges of an ephemeral stream.
The Mallorca sampling sites ranged from pristine to impacted, Magaluf site is in front of a very famous and touristic beach
but it was protected from the "open bay" due to the sensors location behind an island (Isla Sa Porrassa). Sant Elm site is
located in a relatively pristine area near a small harbour, this location includes a sewage plant emissary. Pollença is in an
enclosed bay without high anthropogenic pressure but affected by considerable organic input from the s´Albufereta wetlands,
the emissary of the sewage plant, the Port and the sewer of the urban area. Cap Enderrocat, together with Son Veri and Cala
Blava, are part of a SPAs (*Special Protection Areas*) under the *Birds* Directive and a SIC (site of Community Importance,
Natura 2000 sites) figures that grant a special protection to these areas and count with 11.5% of the Posidonia meadows of the
total flora within the ZEPA Cap Enderrocat- Cap Blanc area. Punta Negra is considered as a Natural Area of Special Interest
(ANEI and a natural space protected by law by the Balearic Islands Government). Sta. Maria, a bay located on the coast of
Cabrera is the most pristine sampling area. Cabrera island is part of a Maritime and Terrestrial National Park located at the



Cabrera Archipelago, and recognized internationally as ZEPA, LIC, Z.E.P.I.M (Special protection zones with importance for the Mediterranean and ZEC (Special zone of conservation). The sampling sites in Mallorca include therefore sites with different degrees of human impact and protected areas with very low anthropogenic impact.


**Figure 1.** Sampling sites included in the study. Locations described in Table 1 and Table 2. (GEBCO 2020)





### 2.1.2 Data analysis

In each site, dissolved oxygen (DO), $pH_{NBS}$, salinity and temperature were measured in both *P. oceanica* and *C. nodosa* meadows with multiparametric sensors (OTT Hydrolab DSX5 and HL4). The duration of the data collection was different
depending on the site, from 1 full day to 4 consecutive days (see Table 1), 24-h periods were used for calculations.

In every station, sensors were deployed 0.2m above the seafloor either in Posidonia or Cymodocea meadows. Data was recorded every 15 minutes except in Cap Enderrocat where readings were taken every 10 minutes (Table 1). Biological metadata detailing habitat, like shoot density and biomass were collected at the time of deployment or collection of the multiparametric sensor.

Sensors were calibrated before each deployment with a two-point pH calibration, with 7.00 and 10.00 NIST traceable pH buffers (Hendriks et al., 2014). Oxygen sensors (Hach LDOTM) were calibrated using the water saturated air method calibration. For validation of salinity, specific conductance calibrations were performed with 50.000uS/cm buffers. For depth measurements, pressure readings were corrected for specific conductance.

Meteorological data of the period during deployment was obtained from the Agencia Estatal de Meteorología (AEMET) for
the stations in Mallorca, and from the Cyprus Department of Meteorology for Cyprus sampling sites locations and from the Hellenic National Meteorological Service for the sampling sites in Crete (see Table A1).


**Table 1.** Characteristics of sampling stations for multiparametric probes. Temperature and salinity are average values during the deployment.

| Region | Station | ID | Days | Species | Depth (m) | Temperature (°C) | Salinity (psu) |
|---|---|---|---|---|---|---|---|
| Mallorca | Cap Enderrocat[1] | 1 | 25/8/2016-26/8/2016 | *Posidonia oceanica* | 14.6 | 26.6 | 40.1 |
| | | | 18/8/2016-29/8/2016 | *Cymodocea nodosa* | 15.7 | 26.6 | 38.8 |
| Mallorca | Son Veri[2] | 2 | 5/06/2012-11/06/2012 | *Posidonia oceanica* | 7.3 | 23.4 | 40.8 |
| | | | 5/06/2012-8/06/2012 | | 5.4 | 23.4 | 40.8 |
| Mallorca | Cala Blava[2] | 3 | 6/06/2012-12/06/2012 | *Posidonia oceanica* | 5.9 | 23.8 | 38.9 |
| | | | 5/6/2012-11/6/2012 | | 4.4 | 23.8 | 39 |



| | | | | | | |
|---|---|---|---|---|---|---|
| Mallorca | Pta Negra[3] | 4 | 2/07/2019-3/07/2019 | *Cymodocea nodosa* | 2.9 | 15.4 | 36.8 |
| | | | 11/04/2019-12/04/2019 | *Posidonia oceanica* | 3.3 | 15.3 | 37.1 |
| Mallorca | Magaluf[2] | 5 | 20/9/2011-23/9/2011 | *Posidonia oceanica* | 6.3 | 26.3 | 40.5 |
| Mallorca | St Elm[2] | 6 | 13/09/2011-16/09/2011 | *Posidonia oceanica* | 9.4 | 26.8 | 40.3 |
| Mallorca | Cabrera[2] | 7 | 6/9/2011-9/9/2011 | *Posidonia oceanica* | 7.2 | 26.6 | 40.2 |
| Mallorca | Pollença[4] | 8 | 16/10/2018-17/10/2018 | *Cymodocea nodosa* | 6.4 | 23 | 38.6 |
| | | | 16/7/2018-17/7/2018 | *Posidonia oceanica* | 6.1 | 24 | 39 |
| | | | 15/1/2019-16/1/2019 | *Posidonia oceanica* | 7.1 | 13.2 | 36.9 |
| | | | 15/1/2019-16/1/2019 | *Cymodocea nodosa* | 7.7 | 13.2 | 37 |
| | | | 18/4/2018-19/4/2018 | *Cymodocea nodosa* | 6.8 | 16.1 | 37.7 |
| | | | 18/4/2018-19/4/2018 | *Posidonia oceanica* | 6.5 | 16.1 | 38.4 |
| | | | 25/06/2015-30/06/2015 | *Cymodocea nodosa* | 8 | 25.7 | 40.6 |
| | | | 25/06/2015-1/7/2015 | *Posidonia oceanica* | 4.5 | 25.8 | 40.9 |
| Crete | Marathi[1] | 9 | 18/7/2017-20/7/2017 | *Posidonia oceanica* | 4.7 | 26.3 | 40.5 |
| | Kalami[1] | 10 | 18/7/2017-20/7/2017 | *Cymodocea nodosa* | 5.4 | 27 | 40 |
| | Maridati[1] | 11 | 21/7/2017-23/7/2017 | *Cymodocea nodosa* | 6.2 | 25.2 | 40.5 |
| | | | 21/7/2017-23/7/2017 | *Posidonia oceanica* | 8.9 | 25.1 | 40.7 |
| Cyprus | Limassol[1] | 12 | 4/09/2017-7/09/2017 | *Cymodocea nodosa* | 3.2 | 27.3 | 40.2 |
| Italy | Revelatta[5] | 13 | 2006-2016 | *Posidonia oceanica* | NA | 18.7 | NA |



| Italy | Revelatta[6] | 13 | 2006-2009 | *Posidonia oceanica* | NA | 18.6 | NA |

**Source:** 1. Unpublished data., 2. (Hendriks et al., 2014)., 3. (Marx et al., submitted)., 4. (Hendriks et al., submitted)., 5. (Champenois et al., 2019)., 6. (Champenois et al., 2012).


## 2.2. Metabolic rate calculations

In order to obtain the metabolic parameters of the studied species, we had to process the profiles obtained with the multiparametric probes (including published and unpublished data described in 2.1).

We calculated the metabolic rates of the seagrass meadows using a modification of the model of Coloso et al. (2008).

implemented in MATLAB (version 7.5. the Mathworks Inc.) explained in detail in Vaquer-Sunyer et al. (2012). As input dissolved oxygen (percent saturation), temperature (° C) and salinity (PSU) from the multiparametric probes was used. For each station, we manually introduced the Mixed Layer Depth (MLD) the latitude, and year, day and time as day fraction. For wind speed (m/s) we used the *k660* calculations based on Kihm et al. (2010).

Net community production (NCP) was calculated as Gross Primary Production (GPP) - Community Respiration (CR), taking

into account diffusive exchange with the atmosphere (D) and other inputs and outputs of DO (A) such as flux between layers and lateral flows following Eq. (1):

$$DO = NCP + D + A. \tag{1}$$

The diffusion with the atmosphere is regulated by the difference in DO concentration linked to atmospheric equilibrium (DO

sat) and the air-sea gas velocity transfer for oxygen *(k)* at a given temperature. Eq. (2).

$$D = k\,(DOsat - DO). \tag{2}$$

where D can be positive (DO addition to the system) or negative (DO removal from the system). Wind speed was estimated at each stations for 15 minutes intervals (10 minutes for the Cap Enderrocat station) to predict *k660* (air-sea gas transfer velocity

for $CO_2$ at 20º C and salinity 35) based on Kihm et al. (2010) and Cole et al. (1998). Schmidt number equations for seawater according to Wanninkhof (1992) were used for the *k* calculation from *k660*. Kihm et al. (2010) proposed three different *k660* estimations: quadratic. cubic. and quartic. These parameterizations cover most of the models for low wind velocities described in the literature. In fact, the cubic model equals the model proposed by Wanninkhof et al. (1999) for short-term winds. Here. the cubic model described by Kihm et al. (2010) is used.

The model assumes that the only metabolic activity during night is respiration (CR) as in the absence of sunlight there is no photosynthetic production. CR can be extracted from the change in $O_2$ concentration during the night (Net Community Production at night = CR), from 1 h past sunset to 1 h before sunrise. During the day light period. Net Community Production (NCP) is considered to be the result of the balance between Gross Primary Production (GPP) and R. NCP was calculated with





the DO rate change within the interval from sunrise to sunset and corrected for other processes. The impossibility of measuring
CR directly during the day caused us to assume that CR during the daytime equals CR during the night (Cole et al., 2000;
Hanson et al., 2003; Lauster et al., 2006). Therefore, GPP can be then estimated adding CR to the daytime NCP, GPP and CR
could be underestimated since it is likely that CR during daytime exceeds CR at night (Grande et al., 1989; Pace et al., 2005;
Pringault et al., 2007) but this underestimation would not affect NCP values (Cole et al., 2000). Individual estimates of CR.,
NCP and GPP within the measured intervals obtained from the multiparametric probes were accumulated over the interval
from sunrise to sunset. The average of these values gave us calculated metabolic rates for each day and station.

As we did not dispose of vertical profiles of Conductivity, Temperature and Depth (CTD) for each station we used the model
of Condie et al. (2001) to calculate the MLD following Eq. (3):

$$S = \frac{\rho C p U_2}{g \alpha H Q}.$$ (3)

where S represents the non-dimensional parameter of the ratio of the input of kinetic energy by the wind to the input of potential
energy by solar radiation. $\rho$ is the density, calculated from the salinity, temperature and pressure collected by the *in situ*
multiparametric sensor following the formula of Fofonoff et al. (1983). $Cp$ represents the specific heat, considered here to be
3850 J kg$^{-1}$ °C$^{-1}$ as the relative value for seawater. $U_2$ refers to the diurnally averaged wind speed specified here to be measured
2 m above the water body  (Simpson et al., 1974; Holloway, 1980). In our case, the wind data was measured at 10 m above
the upstream edge of the water body, So, to convert it, we used the wind profile power law. Eq. (4):

$$\frac{\vartheta}{\vartheta r} = \left(\frac{z}{zr}\right)^{\alpha}.$$ (4)

with $\vartheta$  the wind speed (in meters per second) at a determined height z (metres) and $\vartheta r$ the speed that is known at a reference
height (zr). The exponential $\alpha$ is a coefficient derived empirically which varies upon the stability of the atmosphere. In our
case, neutral stability is assumed and within those conditions $\alpha$ is approximately 0.143. $g$ $(Eq.(3))$, corresponding to the
gravitational acceleration (9.8 m. s$^{-1}$). $\alpha$ $(Eq.(3))$ represents the thermal expansion coefficient which was calculated as a
function of the absolute salinity, *in situ* temperature and pressure. This function is included in the Gibbs-SeaWater (GSW)
Oceanographic Toolbox (McDougall et al., 2011), and evaluates the thermal expansion coefficient α$^t$ in respect with the *in situ*
temperature (t), from equation (2.18.1) of the TEOS-10 Manual (IOC, 2015) following Eq. (5):

$$\alpha^t = \alpha^t(S_{A.}\ t.p) = -\frac{1}{v}\ \frac{\partial \rho}{v \partial T}\Big|_{S_A.}\ p = \frac{1}{v}\ \frac{\partial v}{v \partial T}\Big|\ s_A.p = \frac{g_{Tp}}{g_P}.$$ (5)

This function uses the full TEOS-10 Gibbs function $g(SA.\ t.\ p)$ of (IOC, 2015) as the sum of the (IAPWS, 2008) and (IAPWS,
2009) Gibbs functions.





Finally, H. Eq.(3) equals the average water depth (in m) and Q Eq.(3) the diurnally averaged shortwave radiative heat flux
($Wm^{-2}$).

Then, the surface mixed layer ($z_s$) was approximated following Eq.(6) by Condie et al. (2001):

$$z_s = \left((2.9 - 0.20 \ ln \ S) \pm 0.04\right). \tag{6}$$

To be as accurate as possible, we tried different intervals for the wind and radiation data. Taking in account that stratification
starts to develop between 09:00 and 10:00 hours and reaches its maximum around 16:00 hours (Condie et al., 2001), we
decided to take in account the interval from 9:00 to 16:00 hours for the wind data. This interval resulted to be the more accurate
(highest $R^2$ obtained for that interval. $R^2=0.9$) for the wind data, based on the linear regressions between zH/S obtained with
the different wind and radiation intervals.


### 2.3 Data compilation for benthic chambers

We compiled data from existing literature related to Mediterranean seagrasses metabolism using the benthic chambers
methodology, reaching12 publications based on studies including *P. oceanica* and/or *C. nodosa* meadows. These studies were
carried out from 2000 to 2019. Net Community Production (NCP) was estimated from changes in dissolved oxygen using the
Winkler titration spectrophotometric method (Labasque et al., 2004). The benthic chamber methodology was used to assess
metabolism of seagrass meadows with a total of 100 NCP estimations. In this work we add benthic chambers data to the body
of literature, and we compare the data obtained between both methodologies. We performed ANOVA tests with Rstudio for
the methodological, regional and species comparisons. To analyse the abiotic (wind, pH, depth and salinity) and biotic
parameters (density, shoots and biomass) related to the different metabolic parameters we performed a logistic regression.


**Table 2.** Characteristics of sampling stations for benthic chambers.

| Region | Station | ID | Season | Year | Species | Temperature (ºC) | Salinity (psu) | Depth (m) |
|---|---|---|---|---|---|---|---|---|
| France | Bay of Cavi[1] | 13 | Anual | 1982-1984 | *Posidonia oceanica* | | | |
| Spain | Ebro Delta[2] | 14 | Summer | 2000 | *Cymodocea nodosa* | | | |
| Mallorca | Magalluf[3] | 5 | Anual | 2001 | *Posidonia oceanica* | | | |
| Mallorca | Magalluf[4] | 5 | Summer/Spring | 2001 | *Posidonia oceanica* | 27.5 | | |
| Mallorca | Cabrera[5] | 7 | Summer | 2000 | *Cymodocea nodosa* | 18 | 38 | |
| Mallorca | Sa Paret[5] | 15 | Summer | 2000 | *Posidonia oceanica* | 18 | 38 | |
| Mallorca | Porto Colom[5] | 16 | Summer | 2000 | *Posidonia oceanica* | 18 | 38 | |
| Mallorca | Sta Maria[5] | 7 | Summer | 2000 | *Posidonia oceanica* | 18 | 38 | |
| Mallorca | Magalluf[3] | 5 | Anual | 2002 | *Posidonia oceanica* | | | |



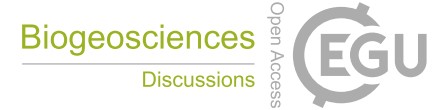

| | | | | | | | | |
|---|---|---|---|---|---|---|---|---|
| Mallorca | Magalluf[3] | 5 | Anual | 2002 | *Posidonia oceanica* | | | |
| Mallorca | Cap Encerrocat[6] | 1 | Anual | 2002 | *Posidonia oceanica* | | | |
| Mallorca | Bay of Palma[6] | 17 | Anual | 2002 | *Posidonia oceanica* | | | |
| Portugal | Ria Formosa[7] | 18 | Summer | 2002 | *Cymodocea nodosa* | | | |
| Greece | Sounion[8] | 19 | Anual | 2006 | *Posidonia oceanica* | | | |
| Mallorca | Es Cargol[9] | 20 | Anual | 2006 | *Posidonia oceanica* | | | |
| Greece | Sounion[8] | 19 | Winter/Spring | 2007 | *Posidonia oceanica* | | | |
| France | Revelatta[10] | 13 | Summer/Winter | 2007-2009 | *Posidonia oceanica* | | | |
| France | Revellata[11] | 13 | Fall | 2012 | *Posidonia oceanica* | | | |
| Mallorca | Alcanada[12] | 21 | Fall | 2012 | *Posidonia oceanica* | 18 | 36.6 | |
| Mallorca | Alcanada[12] | 21 | Winter | 2012 | *Posidonia oceanica* | 13 | 35.94 | |
| Mallorca | Albufera[12] | 22 | Summer | 2012 | *Posidonia oceanica* | 25.5 | 38.27 | |
| Mallorca | Calanova[12] | 23 | Summer | 2012 | *Posidonia oceanica* | 23.5 | 38.14 | |
| Mallorca | Alcanada[12] | 21 | Summer | 2012 | *Posidonia oceanica* | 25.25 | 38.23 | |
| Mallorca | Arenal[12] | 24 | Summer | 2013 | *Posidonia oceanica* | 24.25 | 38.08 | |
| Mallorca | Alcanada[12] | 21 | Spring | 2013 | *Posidonia oceanica* | 18.8 | 37.7 | |
| Mallorca | Arenal[12] | 24 | Summer | 2013 | *Posidonia oceanica* | 27.6 | 37.7 | |
| Mallorca | Calanova[12] | 23 | Summer | 2013 | *Posidonia oceanica* | 28.3 | 37.6 | |
| Mallorca | Albufera[12] | 22 | Summer | 2013 | *Posidonia oceanica* | 23.8 | 38 | |
| Mallorca | Alcanada[12] | 21 | Summer | 2013 | *Posidonia oceanica* | 23.5 | 38 | |
| Mallorca | Pta Negra[13] | 4 | Summer | 2019 | *Posidonia oceanica* | 26.05614 | 37.79036 | 2.793783 |
| Mallorca | Pta Negra[13] | 4 | Spring | 2019 | *Cymodocea nodosa* | 15.85585 | 37.01683 | 5.656805 |
| Mallorca | Pta Negra[13] | 4 | Spring | 2019 | *Cymodocea nodosa* | 15.32636 | 37.42222 | 3.998682 |
| Mallorca | Pta Negra[13] | 4 | Spring | 2019 | *Cymodocea nodosa* | 15.85585 | 37.01683 | 5.656805 |
| Mallorca | Pta Negra[13] | 4 | Summer | 2019 | *Cymodocea nodosa* | 26.18667 | 37.75543 | 2.836086 |
| Mallorca | Pta Negra[13] | 4 | Summer | 2019 | *Cymodocea nodosa* | 26.05614 | 37.79036 | 2.793783 |
| Mallorca | Pta Negra[13] | 4 | Summer | 2019 | *Cymodocea nodosa* | 26.05614 | 37.79036 | 2.793783 |
| Mallorca | Pta Negra[13] | 4 | Summer | 2019 | *Cymodocea nodosa* | 26.18667 | 37.75543 | 2.836086 |
| Mallorca | Pta Negra[13] | 4 | Spring | 2019 | *Posidonia oceanica* | 15.32636 | 37.42222 | 3.998682 |
| Mallorca | Pta Negra[13] | 4 | Spring | 2019 | *Cymodocea nodosa* | 15.32636 | 37.42222 | 3.998682 |
| Mallorca | Pta Negra[13] | 4 | Fall | 2019 | *Posidonia oceanica* | | | |
| Mallorca | Pta Negra[13] | 4 | Summer | 2019 | *Posidonia oceanica* | 26.18667 | 37.75543 | 2.836086 |
| Mallorca | Pta Negra[13] | 4 | Summer | 2019 | *Posidonia oceanica* | 26.18667 | 37.75543 | 2.836086 |
| Mallorca | Pta Negra[13] | 4 | Spring | 2019 | *Posidonia oceanica* | 15.85585 | 37.01683 | 5.656805 |
| Mallorca | Pta Negra[13] | 4 | Spring | 2019 | *Cymodocea nodosa* | 15.85585 | 37.01683 | 5.656805 |
| Mallorca | Pta Negra[13] | 4 | Summer | 2019 | *Posidonia oceanica* | 26.05614 | 37.79036 | 2.793783 |
| Mallorca | Pta Negra[13] | 4 | Spring | 2019 | *Posidonia oceanica* | 15.32636 | 37.42222 | 3.998682 |
| Mallorca | Pta Negra[13] | 4 | Fall | 2019 | *Cymodocea nodosa* | | | |

**Source.** 1.(Frankignoulle et al., 1987), 2.(Barrón et al., 2004).,3.(Barrón et al., 2009)., 4. (Barrón et al., 2006)., 5.(Holmer et al., 2004)., 6. (Gazeau et al., 2005),. 7.(Santos et al., 2004)., 8.(Apostolaki et al., 2010)., 9.(Gacia et al., 2012)., 10.(Champenois et al., 2012)., 11.(Olivé et al., 2016)., 12. (Agawin et al., 2017). 13.(Marx et al., submitted).



# 3 Results

## 3.1 Data compilation and review

We analysed the full dataset, consisting of processed data collected by our research group and additionally the data collected from the literature review with different methodologies and locations in the Mediterranean basin (Data available on repository CSIC) (Fig. 1). In the Western Mediterranean basin, we gathered a total number of 56 observations with sensors for NCP: 34 in summer, five in fall, four in winter and 13 in spring; 50 observations for GPP and CR: 33 for summer, 13 in spring, two during winter and two during fall. In the Eastern Mediterranean, we collected 12 observations with sensors for GPP, NCP and

CR, all of them during the summer. There was only one observation for NCP, GPP and CR during spring for the Eastern Mediterranean using benthic chamber data, this site was sampled every two months during a year (Apostolaki et al., 2010). While in the Western Mediterranean we found a total of 58 GPP and 58 CR measurements using benthic chambers; for *P. oceanica;* 10 observations during fall, nine in spring, 19 in summer and five during the winter, and for *C. nodosa*, 14 observations in summer and one during fall In total, 88 values of NCP were compiled and analysed, 73 of which involved *P.*

*oceanica*: 10 during fall, 16 in spring, 39 during summer and 8 in winter. For *C. nodosa* there were 14 observations during summer and one during fall. There was a total number of 81 NCP estimates of sensors and 100 NCP estimates for benthic chambers.

## 3.3.1 Data analysis

First, we compared results for GPP, CR and NCP between the two studied methodologies: benthic chambers and sensors. We

found significant differences for NCP and GPP ($p<0.001$). (Fig. 2). We therefore analysed data from benthic chambers and from multiparametric probes separately.



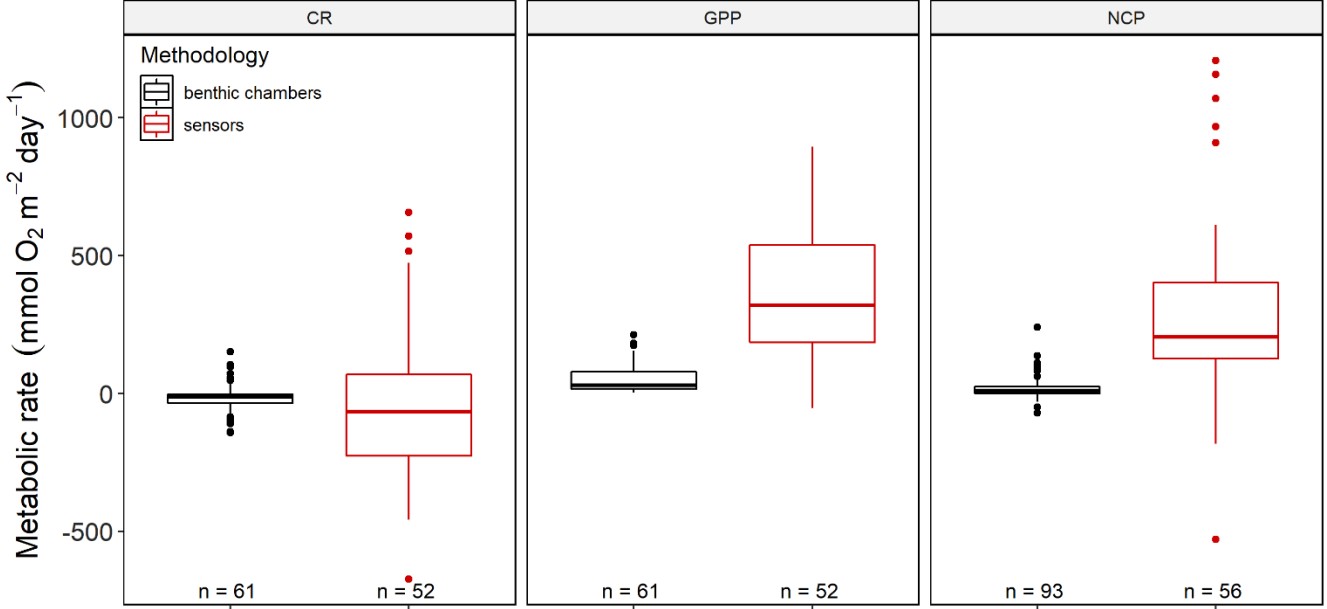

**Figure 2.** Comparison between benthic chambers (black) and sensors (red) for GPP. NCP and CR (mmol $O_2$ m$^{-2}$ day$^{-1}$) for all the dataset. Upper and lower hinges correspond to the upper and lower quartiles, the line inside the boxes correspond to the median and the error bars are based on minimum and maximum standard deviation for each parameter. We found $p<0.5$ for CR. and $p<0.001$ for GPP and NCP.

### 3.3.2 Multiparametric probes

There were no significant differences for any of the three metabolic parameters between the two species (*P. oceanica* and *C. nodosa*): GPP ($p>0.1$), CR ($p>0.1$) and NCP ($p>0.1$). We consequently grouped the sensor data from two species, we found significant differences between Eastern and Western Mediterranean basins for GPP ($p<0.01$) and CR ($p<0.05$) (Fig. 3). The highest GPP rates (Mean $\pm$ SD) occurred during spring with $453.92 \pm 233.3$ mmol $O_2$ m$^{-2}$ day$^{-1}$ and in fall with $241.1 \pm 156.4$ mmol $O_2$ m$^{-2}$ day$^{-1}$, the corresponding CR rates for spring and fall were $61.5\pm379$ mmol $O_2$ m$^{-2}$ day$^{-1}$ and $180.4$ mmol $O_2$ m$^{-2}$ day$^{-1}$  respectively. GPP rates were higher than the corresponding CR rates for all the seasons reflecting that these seagrass meadows tend to be autotrophic ecosystems, reflected in all the positive averaged NCP rates, with the highest values found during spring and summer with $408.08 \pm 454.9$ mmol $O_2$ m$^{-2}$ day$^{-1}$ and $225.2 \pm 280.9$ mmol $O_2$ m$^{-2}$ day$^{-1}$, respectively. We didn´t find any significant results when we compared NCP, CR and GPP between seasons; we therefore pooled the seasons, and we calculated the mean P/R ratio which was above 1 ($1.3 \pm 9.7$), confirming the tendency of net autotrophy. In Addition, for the Western basin monthly and temperature trends for GPP, CR and NCP were studied (See Appendices. Fig. D1). No



significant differences were found for none of the metabolic parameters at a temporal scale and the temperature did not significantly affect any of the metabolic parameters ($p>0.05$).

In the Eastern Mediterranean basin, all data was recorded in summer, with an NCP rate of $349.45 \pm 393.9$ $O_2$ $m^{-2}$ $day^{-1}$; the GPP rate $175.74\pm110.3$ mmol $O_2$ $m^{-2}$ $day^{-1}$ was lower than CR $173.7\pm431.6$ mmol $O_2$ $m^{-2}$ $day^{-1}$, indicating that these seagrass communities tend to be net autotrophic during this period, reflected in an average P/R ratio below 1 ($0.6\pm1.4$). As for the Western basin data, temperature did not have a significant effect on any of the metabolic parameters ($p>0.05$).

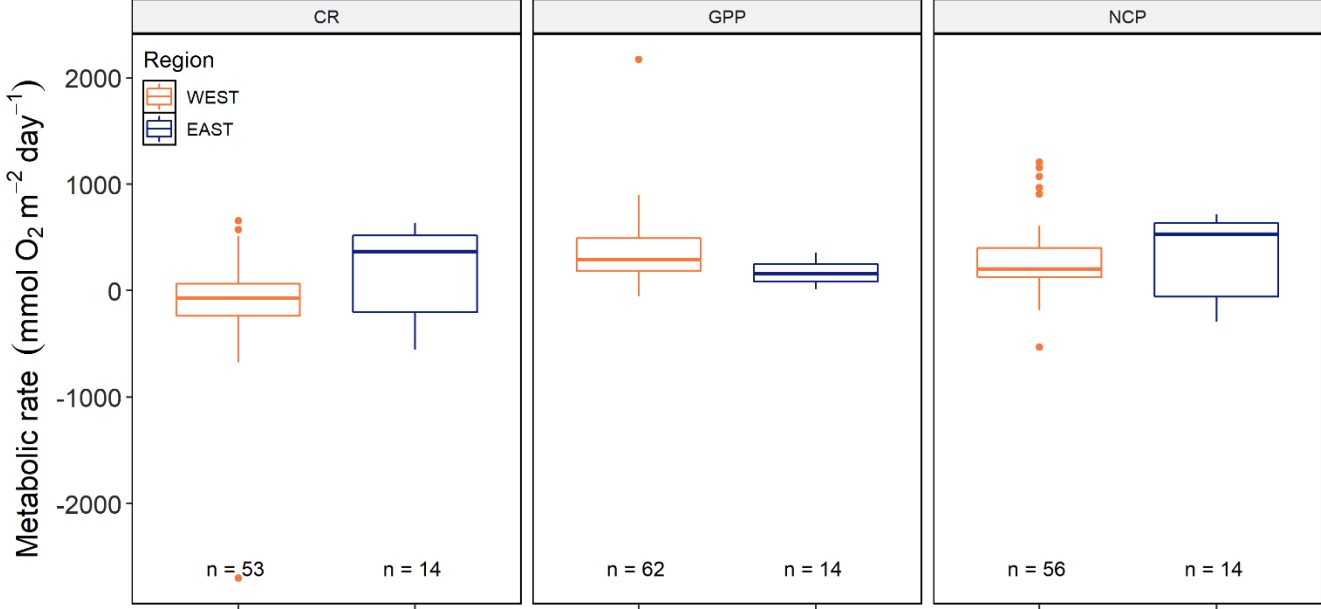

**Figure 3.** Averaged GPP. NCP and CR (mmol $O_2$ $m^{-2}$ $day^{-1}$) values for Eastern (blue) and Western (orange) Mediterranean basins from multiparametric sensors dataset. Upper and lower hinges correspond to the upper and lower quartiles, the lines inside the boxes correspond to the median and the error bars are based on minimum and maximum standard deviation for each parameter. We found p<0.05 for GPP and CR.

The measurements in the Eastern and Western Mediterranean basins showed significant differences in the CR (p<0.05) and GPP ($p<0.05$). (Fig. 3). During summer, the highest averaged GPP in the Western basin was $483.10 \pm 705.3$ mmol $O_2$ $m^{-2}$ $day^{-1}$, more than two times higher than the averaged GPP in the Eastern basin ($175.74\pm110.3$ mmol $O_2$ $m^{-2}$ $day^{-1}$). Averaged NCP values for the two regions were positive, showing that the ecosystems tend to be net autotrophic and therefore act as carbon sinks. NCP in the Eastern basin was $349.45 \pm 393.9$ mmol $O_2$ $m^{-2}$ $day^{-1}$ which was higher but not statistically different ($p>0.1$) from the Western basin ($225.2\pm 280.9$ mmol $O_2$ $m^{-2}$ $day^{-1}$). The high SD values may be due to the high variability found in the individual values. During summer, NCP in the Eastern basin ranged from -293.7 mmol $O_2$ $m^{-2}$ $day^{-1}$ to 713.6 mmol $O_2$ $m^{-2}$ $day^{-1}$ and fluctuated from 23.5 to 1207.4 mmol O2 $m^{-2}$ $day^{-1}$ in the Western basin. The temperature recorded during the highest NCP measurement in the Western basin was 26.6ºC, which was not the highest temperature recorded, and close, even though




a bit higher, to the optimal value reported for *P. oceanica* of 25.8 ℃ (Savva et al., 2018). For the Eastern Mediterranean basin, the highest GPP obtained was 357.31 mmol $O_2$ m$^{-2}$ day$^{-1}$ at Limassol station (Cyprus) during September and the *in situ* temperature registered at that moment was 27.7℃, which was not the highest temperature registered in the Eastern basin (28.5℃) but higher than the mean temperature in the Eastern basin during the summer sampling campaign (25.9±0.8 ℃). Lowest GPP values found in the Western and Eastern regions were different, we found a negative GPP of 3.81 mmol $O_2$ m$^{-2}$ day$^{-1}$ for the Western basin in the Cala Blava station (Mallorca) during spring whereas the lowest GPP value in the Eastern basin was 14.12 mmol $O_2$ m$^{-2}$ day$^{-1}$ in Marathi station (Crete) in summer; temperatures during both measurements were similar with less than one Celsius degree of difference between them (26.7℃ in Marathi station (Crete) and 25.9℃ in Pollença station (Mallorca).

### 3.3.3 Benthic chambers

We found significant differences for NCP ($p<0.001$) and GPP ($p<0.001$) (Fig. 4) between *P. oceanica* and *C. nodosa* in the Western Mediterranean basin. We therefore examined those two species separately. As we didn´t have *C. nodosa* data for the Eastern Mediterranean basin we only examined *P. oceanica* to distil patterns between Eastern and Western Mediterranean basin regions. There were significant differences for NCP ($p<0.05$). GPP ($p<0.1$) and CR ($p<0.05$) for *Posidonia* between Eastern and Western regions (Fig. 5). At a seasonal scale, there were no significant differences for NCP, GPP or CR for *C. nodosa* in the Western basin ($p>0.05$), even if there were identifiable trends between seasons. Except for the summer. Production was lower than respiration during fall and spring, this was reflected in the averaged NCP, with a negative rate (-9.2 ±23.0 mmol $O_2$ m$^{-2}$ day$^{-1}$), revealing that the *C. nodosa* community tend to be net heterotrophic, also reflected in the averaged P/R ratio below 1 (-1.05±1.8). For *P. oceanica* seasonal metabolic trends were also studied in the Eastern Mediterranean basin and the Western Mediterranean basin. There were no significant results for the different seasons between NCP, GPP and CR for the Eastern or Western basin so we pooled the seasons for both regions. For the Western basin, averaged NCP was (19.62 ±28.2 mmol $O_2$ m$^{-2}$ day$^{-1}$). GPP rate (66.562±28.2 mmol $O_2$ m$^{-2}$ day$^{-1}$) was higher than the CR rate (-13.9±57.4 mmol $O_2$ m$^{-2}$ day$^{-1}$) which reflect the tendency of *P. oceanica* communities to be net autotrophic. Additionally, we examined monthly NCP, CR and GPP for *C. nodosa* in the Western basin and their variability with temperature. We repeated the same analysis for *P. oceanica* in the Western basin (See Appendices Fig. D1, E1). There were no statistical differences in a temporary scale for none of the species, regions, or metabolic rates evaluated. Nevertheless, we found remarkable differences in individual values for all of them between the years for each species. For *C. nodosa* in the Western basin. NCP values ranged from -71.46 mmol $O_2$ m$^{-2}$ day$^{-1}$ during summer 2000 to 34.02 mmol $O_2$ m$^{-2}$ day$^{-1}$ during the summer of 2001. The lowest respiration rates were found in summer 2000 with an individual CR value of -83.65 mmol $O_2$ m$^{-2}$ day$^{-1}$ and the highest rate reached 104.51 mmol $O_2$ m$^{-2}$ day$^{-1}$ in summer 2002.



As more data was available for *P. oceanica*, we were able to analyse its metabolic rates regionally (Eastern and Western
Mediterranean basins) and temporally (seasonally, monthly, and yearly). For the East region, we found the highest *P. oceanica*
individual NCP value during spring with a metabolic rate of 63.85 mmol $O_2$ m$^{-2}$ day$^{-1}$ and the lowest was found during fall with
-106.64 mmol $O_2$ m$^{-2}$ day$^{-1}$. Regarding CR, during summer the highest value was -58.11 mmol $O_2$ m$^{-2}$ day$^{-1}$ and the lowest was
-106.64 mmol $O_2$ m$^{-2}$ day$^{-1}$. About the Western region, where the higher amount of data was available, we found a maximum
NCP for *P. oceanica* of 136.85 mmol $O_2$ m$^{-2}$ day$^{-1}$ during summer 2001 and a minimum value of -15.4 mmol $O_2$ m$^{-2}$ day$^{-1}$
during same summer. For the CR in this region for Posidonia, we found values ranging from -141.9 mmol $O_2$ m$^{-2}$ day$^{-1}$ in
summer to 150.8 mmol $O_2$ m$^{-2}$ day$^{-1}$ in fall.

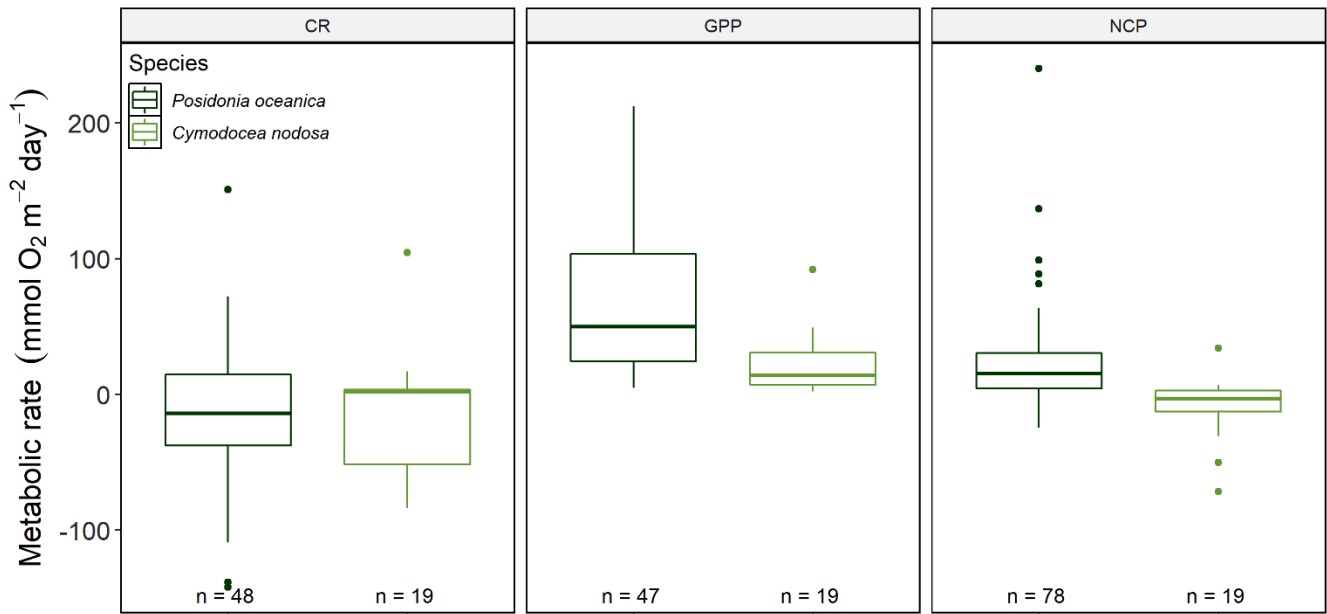

**Figure 4.** Differences for GPP. CR and NCP (mmol $O_2$ m$^{-2}$ day$^{-1}$) between *Cymodocea nodosa (*light green) and *Posidonia oceanica* (dark
green) for the benthic chambers dataset publications in the Western Mediterranean basin. Upper and lower hinges correspond to the upper
and lower quartiles, the line inside the boxes correspond to the median and the error bars are based on minimum and maximum standard
deviation for each parameter ($p$<0.001 for GPP and NCP).





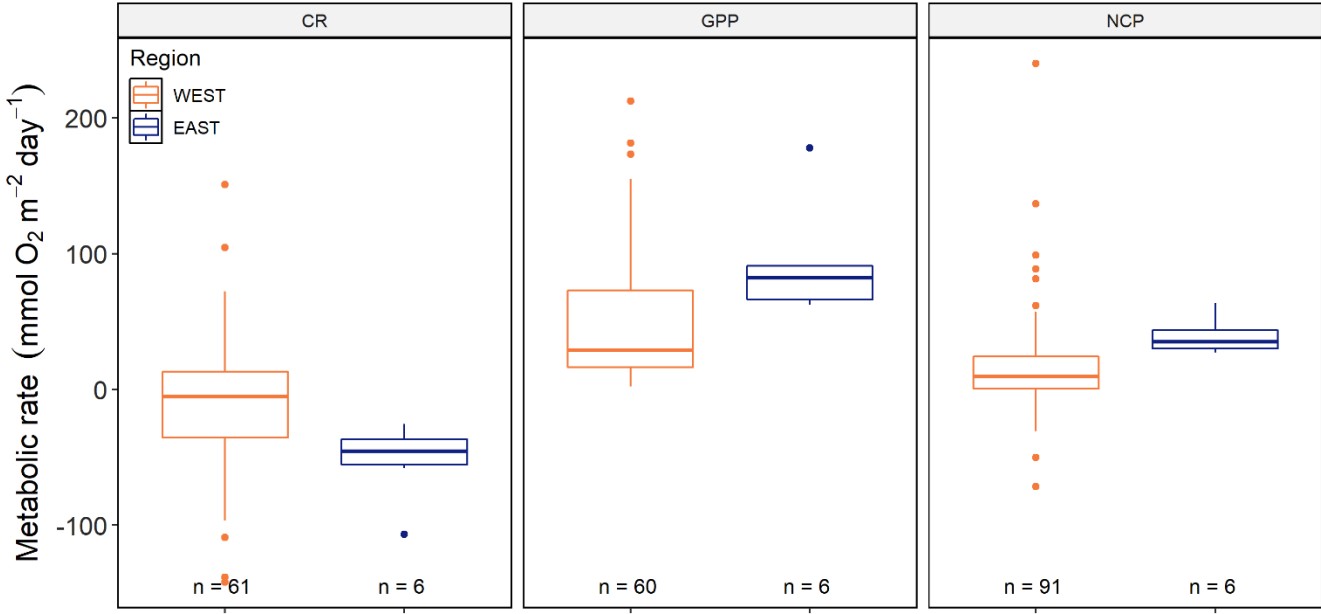


**Figure 5.** Differences in GPP. NCP and CR (mmol $O_2$ m$^{-2}$ day$^{-1}$) between Eastern (blue) and Western (orange) with the benthic chamber dataset for *Posidonia oceanica*. Upper and lower hinges correspond to the upper and lower quartiles. the line inside the boxes correspond to the median and the error bars are based on minimum and maximum standard deviation for each parameter ($p<0.05$ for NCP and CR and $p<0.1$ for GPP).


**4 Discussion**

**4.1 Sensors**

We didn´t detect differences in the metabolic rates between *Cymodocea nodosa* and *Posidonia oceanica*, probably due to the effect of lateral advection and mixing, since the sensors are measuring in the water column, and even though they are within

the meadow at 0.2 m of the seafloor. There apparently is enough mixing and transport of water between the seagrass meadows of the two species to appreciate the difference. The studied *C. nodosa* and *P. oceanica* meadows were only separated by a distance of less than 10 m, which may add to the lack of differentiation of values found between the two studied species. Measurements with multiparametric probes should therefore be interpreted as an ecosystem level measurement and not as



values at species level. The influence of phytoplankton and other primary producers may affect sensors and benthic chamber´s measurements. A relationship between Chl *a* in the water column and measured GPP during the spring phytoplankton bloom in the Bay of Palma was demonstrated (Gazeau et al., 2005), but in a study taking into account annual patterns in the Bay of Revellata (Italy), Champenois et al. (2012) reported that the highest GPP values of *P. oceanica* where found at a time when planktonic Chl *a* was particularly low and the highest values of Chl *a* didn´t reflect an increase in GPP and GPP values where thus uncorrelated with planktonic Chl *a*. Pooling measurements in both seagrass species allowed us to estimate the metabolic

activity of the whole ecosystem and compare between regions. Our results showed significant differences for CR and GPP between the Eastern and Western Mediterranean basins. The presence of positive CR values may be due to lateral advection of a water mass with a higher oxygen content or may reflect the presence of a physical event in the moment of the measurement as upwelling. Without these individual positive CR values, CR in the Eastern basin would have averaged -429.0 ± 138.5 mmol $O_2$ m$^{-2}$ day$^{-1}$, which would have been two times lower than the averaged CR rate in the Western basin (-230.3 ± 137.5 mmol $O_2$

m$^{-2}$ day$^{-1}$). The positive averaged NCP values for both regions illustrate the trend of those ecosystems to be net autotrophic and act as carbon sinks.

The highest Gross Primary Production measured was in Cap Enderrocat (Mallorca. Spain) during summer and reached 2169.61 mmol $O_2$ m$^{-2}$ day$^{-1}$, this is the highest GPP value ever reported before in a seagrass meadow, higher than the 1338.0 mmol $O_2$ m$^{-2}$ day$^{-1}$ at the bay of Revellata (Italy) registered by Champenois et al. (2012). In this study, Champenois et al. (2012) infer

that extreme GPP values in *P. oceanica* meadows may be rare events that are hardly captured by the classic benthic chambers methodology. Nevertheless, the presence of high values in our study might reflect that these events could be less uncommon than previously thought. Looking at the compiled data, trends related to the temperature appeared (Fig. D1). The highest CR values as well as GPP values matched with the highest temperatures, confirming that increasing temperatures enhances respiration (Brown et al., 2004). Abiotic and biotic factors that drive seagrass community metabolism differ between regions.

The Eastern basin data showed that GPP was affected by temperature ($p<0.05$). For the Western basin data, we clearly found that depth affected GPP ($p<0.05$), as depth determines light availability which in turn determines seagrass distribution., biomass and productivity (Dennison, 1987). The lack of significance of depth affecting GPP in the Eastern basin may reflect the depth homogeneity between the sampling locations (Table 1). In both the Eastern and Western basins, wind was a factor driving NCP ($p<0.05$), this reinforces our hypothesis that mixing of the water column, together with the lateral advection affects

measurements of multiparametric sensors. Also, CR was correlated with windspeed ($p<0.005$) in the Eastern and Western basin (Fig. C1). Additionally, in the Western basin CR was correlated with depth ($p<0.001$) (Fig. B1). Biotic parameters like shoot density and biomass were not determinant for GPP, CR nor NCP ($p>0.1$), which underlines the effect of mixing of water for multiparametric probes.





**4.2 Benthic chambers**

With benthic chambers data, the distinction of differences in productivity between *Posidonia oceanica* and *Cymodocea nodosa* (Fig. 4) was possible. This distinction between species includes epiphyte and bacterial communities associated to each seagrass species. Posidonia had higher biomass and was more productive compared to Cymodocea. However, we could only analyse these differences in the Western Mediterranean basin as data for Cymodocea from the Eastern basin wasn´t found in the
bibliographic research. When we compared the two species, we found that Posidonia was more productive than Cymodocea (Fig.4), this may be due to the highest number of Posidonia GPP values (n=42), compared to the Cymodocea ones (n=19), together with the fact that the mean GPP for Posidonia within the published data was higher than the GPP obtained from the analysed profiles (74.3±55.4 and 22.2±6.7 mmol $O_2$ m$^{-2}$ day$^{-1}$. respectively).

We found regional differences for *P. oceanica* NCP ($p<0.050$) between Eastern and Western Mediterranean basins. The
absence of significant results for the rest of metabolic parameters (GPP and CR) may be due to the fact that there was not enough data available for the Eastern region as only the values provided by the study of Apostolaki et al. (2010) were available. After the compilation of the published data, we found remarkable differences between the two areas: around 42 individual values were available for the Western basin while only six data for the Eastern region. Even that temperature remains the same than for the water column, no significant effect of abiotic parameters on seagrass metabolism was found. This lack of
significance may be due to the fact that seagrasses in benthic chamber experiments are isolated and water renewal is limited (Champenois et al., 2012, 2019). At a seasonal scale, no significant results were found for *P. oceanica* whereas for *C. nodosa* NCP during summer was different from spring and fall campaigns. The absence of significant results for *P. oceanica* may reflect the lack of data during other periods of the year. Nevertheless, there were remarkable individual differences at species, regional and temporal scales. A clear higher production of *Posidonia oceanica* was observed during the spring/summer months
(Fig. E1) that reinforces the previously higher production described in previous publications. For both species, GPP and NCP values obtained through the profile analysis were consistently lower than literature ones. Thus, we evidenced a diminution of the highest GPP and NCP values compared to data published in previous years. The differences of metabolic rates during the same season seems related to different factors such as the volumes of the benthic chambers which affects the value of the final GPP (Champenois et al., 2019) and the biomass of the seagrass studied. Despite the non-significant results of the metabolism
between seasons, a clear variation of GPP, NCP and CR with temperature is observed for both species (Fig. D1). Patterns show the same trends that the previously described for multiparametric probes. Even if some of the individual NCP values were negative, the average NCP for *P.oceanica* and *C.nodosa* was higher than 0. We therefore considered these communities as net autotrophic during the sampling periods proving that *P. oceanica* meadows were more productive than the *C. nodosa* meadows, in agreement with previous studies (Duarte et al., 2010; Champenois et al., 2012, 2019).






### 4.3 Sensors vs Benthic chambers

When we compared methodologies, we found significant differences for GPP, NCP and CR. Indeed, the sensor data values were found four orders of magnitude higher compared to the benthic chambers, as previously reported by Champenois et al. (2012). This difference may be due to a possible underestimation of the metabolism rates assessed by the benthic chambers methodology. There are some limitations linked to benthic chambers methodology as even if the seagrass is submerged in the sea water, there is no interchange with the water column. and nutrient limitation could occur. Nutrient assimilation in seagrasses is mostly done by the leaves in the water column (Alcoverro et al., 2001), without renewal of nutrients in the water column, this assimilation decreases and its negative effect intensifies as the incubation time increases, affecting therefore measurements of seagrass metabolism and obtained values that could, consequently, be lower due to this limiting factor. Nevertheless, there is no oxygen or nutrient limitation when incubations are short (24h) (Barrón et al., 2009). Another possible explanation for the underestimation in benthic chambers could be the fact that roots and rhizomes may be cut by the PVC base of the benthic chambers ring even though this should be considered a rare event as rhizomes are usually put down so the roots are not cut. For the seagrass physiology, rhizomes play an important role as they translocate resources between shoots (Marbà et al., 2002), affecting therefore seagrass metabolism if they are severed. Another reason that may explain this underestimation in the metabolism values is the fact that pH may increase, together with $O_2$ during the day; those two factors, together with a high irradiance conducts the Ribulose-1.5. biphosphate-carboxylase-oxygenase enzyme to change from carboxylase to oxygenase (Heber et al., 1996). Under this reaction there is a higher consumption of $O_2$ and a $CO_2$ exudation which may conduce to a lower GPP estimation from the change in $O_2$ (Champenois et al., 2012). An additional hypothesis is the reduction of the width of the diffusive boundary layer (DBL) between a seagrass leaf and the water column due to the reduction in motion and water exchange with the water column, since one of the factors that determines DBL boundary thickness is the water velocity (Enríquez et al., 2006; Hendriks et al., 2017). Therefore, incubations using benthic chambers can possibly underestimate GPP values while multiparametric probes could probably provide more realistic GPP values. Nonetheless, benthic chambers provide a more species-specific measurement as they are not affected by the surrounding photosynthetic organisms whereas multiparametric probes metabolism values reflect an ecosystem estimation. Compared to the eddy covariance methodology, benthic chamber metabolic estimations are lower (Koopmans et al., 2020), in their study, Koopmans et al. (2020) have reported a NCP for *P. oceanica* ranging from 54 to 119 mmol m$^{-2}$ d$^{-1}$, a rate lower than the estimates from multiparametric probes. The chosen method should therefore be selected depending on the study objectives taking in account the factors mentioned. Independent of the different methodologies, we confirm two main limitations: the lower amount of available data within the Eastern Mediterranean and the higher sampling frequency during summer compared to other seasons. Within the analysed multiparametric sensor data, we had 33 individual values for the Western basin and only 8 for the Eastern basin. At a seasonal level, 70.5% of data was sampled during summer, 19.2% during spring, 7.3% during fall and 3% during winter. This pattern is clearly repeated in the published data. The analysed chamber data was more balanced seasonally with the same amount of observations for summer fall and spring, due to the experimental design of the studies but we had no data for the Eastern basin.





We also highlight the lack of data for *C. nodosa* compared to *P. oceanica*. This unbalance between regions and seasons may
have biased our analysis and could be the reason why some of the abiotic factors didn´t significantly reflect its influence on
the metabolism, together with the absence of significance within seasonal results. For future studies, in order to better evaluate
both Western and Eastern Mediterranean basins and to compare between them sampling locations in the Eastern region should
be increased and a more data should be collected during all the seasons, specifically in winter, which is the period with less
available data.

515

520

### 4.4 Seagrass metabolism and carbon burial

The multiparametric probes dataset in the Western region reflected that 80.3% of the NCP values were positive. This
percentage was even higher for the benthic chambers in the same region with 86.7% of positive NCP values. These high
percentage of NCP values reflects the strong capacity of the seagrass meadows to act as carbon sinks, which is also exemplified
by the P/R ratio above one for both methodologies. Therefore, *C. nodosa* and *P. oceanica* communities tended to be net
autotrophic with higher productive rates shown for *P. oceanica*. The exception for this net autotrophic averaged NCP values
was found in Maridati station where the averaged NCP was -236.1 mmol $O_2$ m$^{-2}$ day$^{-1}$ during July 2017. This station is located
nearby a temporary stream and receive its discharges which implies an extra nutrient input into this area. This input may have
caused physiological stress to the seagrass meadow and caused it to be net heterotrophic during this period. We also found
negative individual values in NCP for Pollença during spring 2018 (-137.1 mmol $O_2$ m$^{-2}$ day$^{-1}$) and for Cap Enderrocat (-528.8
mmol $O_2$ m$^{-2}$ day$^{-1}$ ) sites in summer 2016. Both sampling sites were located near by a nutrient source input: a torrent in the
case of Cap Enderrocat and organic inputs coming from s´Albufereta coastal lagoon in the case of Pollença. Additionally, in
the case of Pollença site, the organic matter inputs coming from the Port and the sewage systems from urban areas and hotels
may have affected the metabolism of the seagrasses in this site. The negative NCP values corresponded to sampling sites
located nearby organic matter sources. The bigger input of nutrients from the organic matter sources is reflected in the data
collected, showing the impact that might have on the seagrass meadows (Borges et al., 2013). For future studies, we suggest
comparing water nutrient values during the measurements with the NCP output. Unlike in Maridati, we did have positive NCP
averaged values for Pollença and Cap Enderrocat, an illustration of how those communities can switch from autotrophy to
heterotrophy depending on the physiological stress due to anthropogenic impacts. These ecosystems are mainly net autotrophic
and hence act as carbon sinks but might be threatened and disappear due to high organic inputs. Both evaluated methodologies
can be very useful tools to monitor the health of vegetated marine ecosystems. Benthic chambers can be very suitable if the
evaluation of a specific species is needed while the use of multiparametric probes is a very convenient, robust, and an easy to
manage tool in order to assess seagrass metabolism at a community scale. This seagrass ecosystem monitoring methodologies
could therefore be relevant tools for the prevention and conservation of those invaluable ecosystems.




## 5 Conclusions

The assessment of seagrass metabolism and obtained ranges for Net Community Production, Gross Primary Production and Community Respiration are significantly different depending on the methodology used. Ranges obtained with benthic chambers are lower compared to the values obtained with multiparametric sensors. The benthic chamber methodology allows

the evaluation of seagrass metabolism at a species level, and significant differences between *Posidonia oceanica* and *Cymodocea nodosa* for GPP and NCP were observed, with *P.oceanica* being the more productive species compared to *C.nodosa*. Multiparametric sensors can assess metabolism at a community/ecosystem level of the system. Despite these differences, both methodologies exposed significant differences between the Eastern and Western Mediterranean basins, in GPP and CR for the multiparametric sensors and for CR, GPP and NCP in the benthic chambers. In addition, there is a clear

pattern between the seagrass metabolism and temperature. There is a publication bias leading to a higher number of observations in the Western region and a more elevated number of observations for summer compared to other seasons. The longer observation period allowed by the use of multiparametric sensors compared to benthic chambers, allowed us to describe high GPP values not previously reported in the literature. Furthermore, the sampling during different time periods revealed the switch between negative and positive NCP values, showing that the autotrophy or heterotrophy status can change in a same

location during different periods of the year, highlighting the importance of monitoring during the year and not only summer. The analysed data, in agreement with the published data show that *P. oceanica* and *C. nodosa* communities are net autotrophic in almost all the seasons and locations sampled stressing their key role for climate change mitigation, by acting as carbon sinks. Therefore, it is important to keep studying the evolution of seagrass metabolism in order to have further knowledge about the state of those ecosystems and to prevent their deterioration in a climate change context where they play an essential role.








## 6 Appendices

**Appendix A.** Sampling sites coordinates and related Meteorological stations.

| Sampling station | Latitude (°) | Longitude (°) | Meteorological station | Latitude (°) | Longitude (°) |
|---|---|---|---|---|---|
| Cap Enderrocat (Mallorca) | 39.473 | 2.721 | Palma Son San Juan (Mallorca) | 39.561 | 2.737 |
| Son Veri (Mallorca) | 39.495 | 2.73 | | | |
| Cala Blava (Mallorca) | 39.489 | 2.724 | | | |
| Pta.Negra | 39.552 | 2.61 | | | |
| Magaluf (Mallorca) | 39.537 | 2.674 | Palma CTM (Mallorca) | 39.553 | 2.625 |
| St. Elm (Mallorca) | 39.726 | 2.603 | | | |
| Sta. Maria (Mallorca) | 39.15 | 2.96 | Pollença (Mallorca) | 39.909 | 3.1 |
| Pollença (Mallorca) | 39.826 | 3.088 | | | |
| Marathi (Crete) | 35.504 | 24.174 | Chania (Crete) | 35.553 | 24.068 |
| Kalami (Crete) | 35.47 | 24.136 | | | |
| Maridati (Crete) | 35.222 | 26.273 | Sitia (Crete) | 35.205 | 26.095 |
| Limmassol (Cyprus) | 34.707 | 33.123 | 1389-7615 Tepak (Cyprus) | 34.677 | 3.038 |

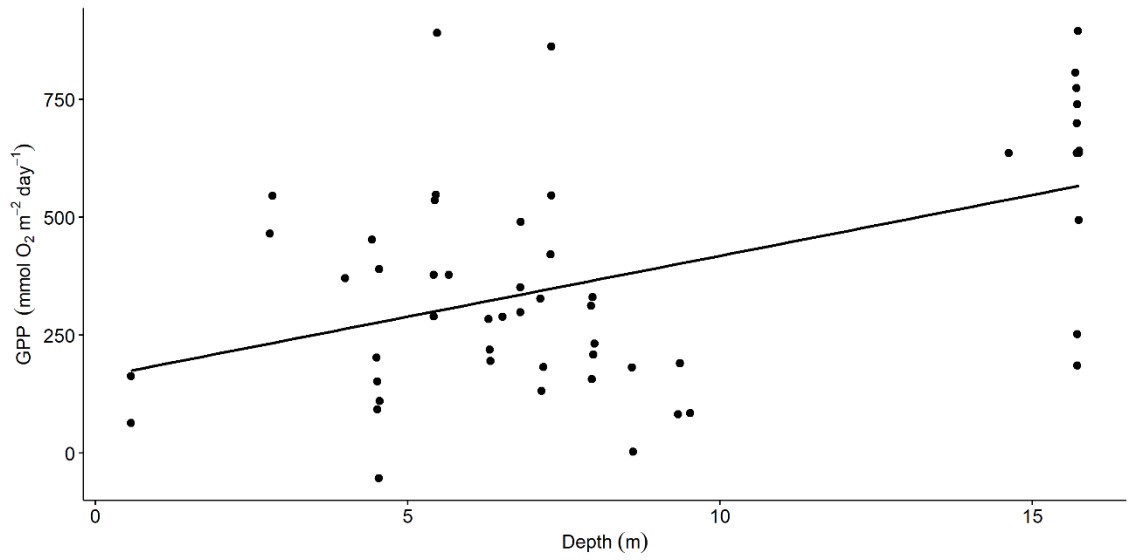

**Appendix B.** Significant abiotic factors related to out Western basin multiparametric probes data. (A) Depth (m) correlation with GPP (mmol $O_2$ m$^{-2}$ day$^{-1}$) with r$^2$=0.45 and p<0.001.



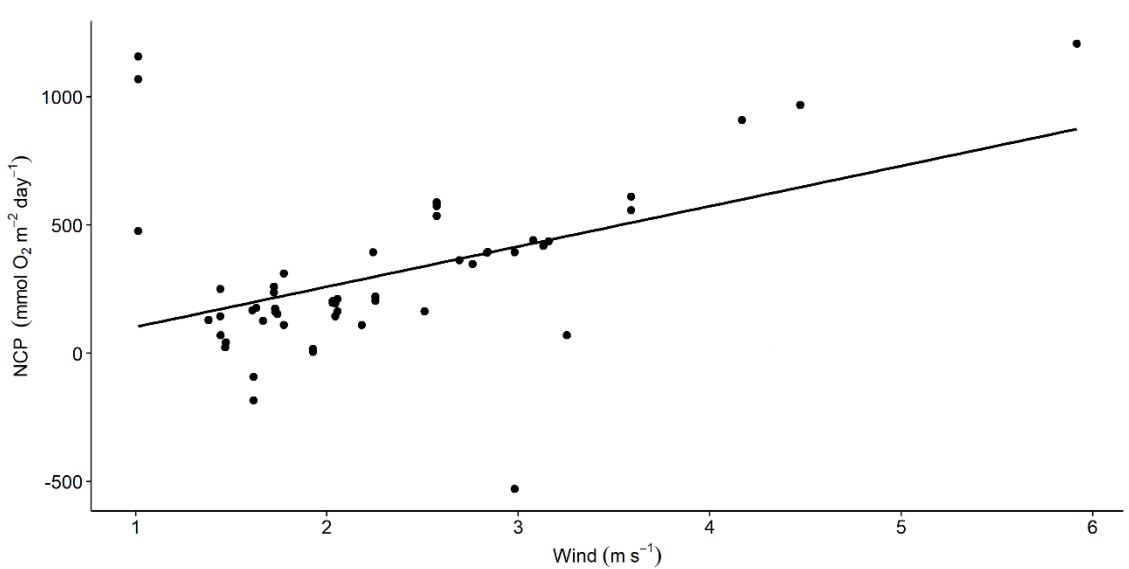

**Appendix C.** Significant abiotic factors related to out Western basin multiparametric probes data. Wind (m/s) correlation with NCP (mmol $O_2$ m$^{-2}$ day$^{-1}$). With r$^2$=0.45 and p<0.001.


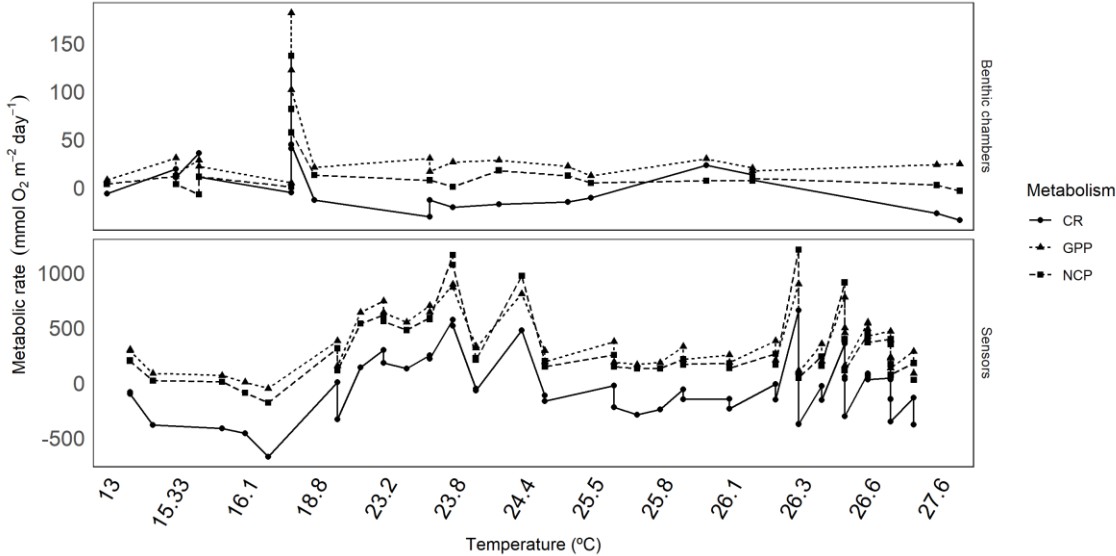

**Appendix D.** CR, NCP and GPP (mmol $O_2$ m$^{-2}$ day$^{-1}$) variation with the in situ temperature in the sensors of the Western basin with all the compiled data (bottom plot) and in the *Posidonia Oceanica* benthic chambers of the Western basin with all the compiled data (top plot).






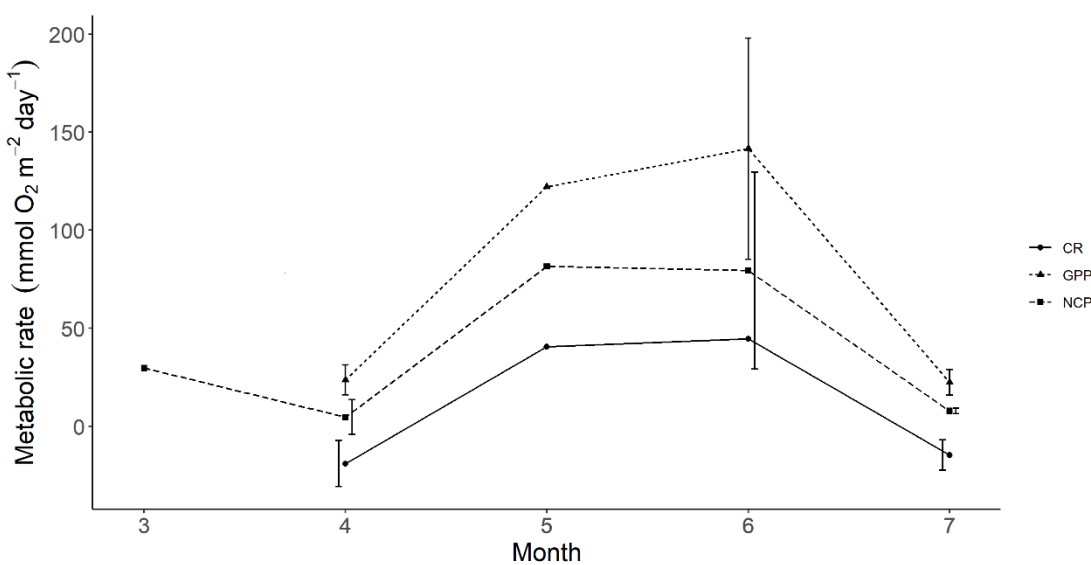

**Appendix E.** *Posidonia oceanica* monthly GPP, NCP and CR (mmol $O_2$ m$^{-2}$ day$^{-1}$) trends in Western Mediterranean basin benthic chambers
with our benthic chambers data, together with literature data. Error bars are based on minimum and maximum standard deviation for each
parameter.



**Code availability**

Metabolic rates of the seagrass meadows were calculated using a modification of the model of Cole et al. (2000) , implemented

in MATLAB (version 7.5, the Mathworks Inc.) explained in detail in Vaquer-Sunyer et al. (2012)



**Data availability**

We want firmly our data to be publicly available. We are working to upload it on a suitable repository.

**Executable research compendium (ERC)**


**Author contribution**

Conceptual idea IEH and NM. Data collection MW. SF. RVS. NM. analyses AEM. SF. IH. literature compilation AEM. all authors have contributed to the writing of the article.

**Competing interests**

The authors declare that they have no conflict of interest.

**Acknowledgements**

This work was funded by the Spanish Ministry of Economy and Competitiveness (Project MEDSHIFT, CGL2015-71809-P) and Project RTI2018-095441-B-C21 (SUMAECO) from the Spanish Ministry of Science, Universities and Innovation. We would like to also thank E. Apostolaki for her reviews and comments.





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
