# Peer review of "Mediterranean seagrasses as carbon sinks: Methodological and regional differences"

_Biogeosciences, 2021_

## Author Comment (AC1)

**Reply to Comment on bg-2021-60**

**Referee #1**

The manuscript by Escolano-Moltó et al. presents a synthesis of seagrass metabolic data from previously published work and/or datasets in the Mediterranean relative to two seagrass species (*Posidonia oceanica* and *Cymodocea nodosa*) using two methodologies (benthic chambers and multiparametric sensors). This is a very relevant topic in the current context of climate change in relation to carbon sequestration in coastal areas, and the work presented has a considerable amount of data and results that fit within the scope of Biogeosciences. While the seagrass metabolic data is not particularly novel, the comparisons among methods, species, and regions (Mediterranean basin) are very important. However, there is a major flaw in the statistical approach used and how this is used to pooling datasets. As presented in the manuscript, the ANOVA analysis is not considering the lack of independence in the data from the same season, site, or region and should be reviewed. Depth should also be considered as a covariate as it is most likely related to the metabolic rates due to the light availability. Increasing the accuracy in the statistics presented is essential for the interpretation of the results presented here, especially because datasets are pooled based on those analyses and then further analyses are done. Therefore, the results presented are built over potentially incorrect statistical analysis, and, right now, it is not possible to evaluate the accuracy of the entire set of results presented. If ANOVA assumptions cannot be met, consider using a different statistical approach (e.g. mixed models) and present the results accordingly. Especially critical is the pooling of datasets, if possible, this should be avoided and instead, grouping factors or separate analysis should be considered. Additionally, the main text structure needs revision (see specific comments below). In particular, there is a lot of information on the methods section that is missing in the Results (e.g. habitat traits measured, logistic regressions between abiotic and biotic parameters, pH data). Also, there are Results (including stats) presented in the Discussion section. Throughout the text, there are several typos and constant misuse of species names, which appear sometimes complete and others shortened, and many times italics are not used. I believe the work presents interesting data, and so, the analyses could be revised to improve the way results are presented and discussed in the manuscript. Hopefully, my suggestions help to improve the manuscript. All my comments are made with this purpose.

Reply: We thank the referee for the helpful comments, we have restructured the text as suggested, and taken all the specific comments into account. We understand the concerns about the ANOVA analyses and have redone the analyses using mixed models and included depth as a factor. See replies to the specific comments below.

**Abstract:**

L14. I would recommend replacing ": "Through their metabolic activity, they ..."
with "Seagrasses". As it is written now, the statement neglects the fact that
carbon stored in sediments can come from external sources and that the
buffer of low pH can also occur due to other processes not related to the
seagrass aerobic metabolism.

        Reply: Thank you for the suggestion, we have modified the text
accordingly. The sentence now reads: "Seagrasses can act as carbon sinks;
buffer lowering pH values during the day and store carbon in the sediment
underneath their meadows."

L15. This is a long sentence that could be re-written to increase clarity. For
instance: In this study, we analysed published and own (unpublished?) data on
seagrass community metabolism to evaluate trends through time of these two
species comparing two methodologies: benthic chambers and multiparametric
sensors.

        Reply: Thank you for the suggestion. The modification has been included
in the manuscript. The sentence now reads: "In this study, we analysed
published and previously unpublished own data on seagrass community
metabolism to evaluate trends through time of these two species comparing
two methodologies: benthic chambers and multiparametric sensors."

L19. remove "with no significant results despite the clear visual trends."

        Reply: Modified in the text.

L21. Add a comma before whereas

        Reply: Added to the text.

L23. add "the" before highest or replace by higher

        Reply: Added to the text.

L23 - L24. write the complete species name in italics and remove the genus (i.e. *P.oceanica*, *C. nodosa*)

> Reply: This was modified in the text.

**Introduction**

General comment: The introduction is long, there is a lot of information and it is difficult to follow the flow of ideas. This is especially the case around the importance of seagrass aerobic metabolism related to (1) carbon burial in sediments and (2) buffering of low pH. Both processes are related to primary productivity, however, there are differences among them that right now are unclear in the text. I would recommend reviewing the text, try to shorten it, and present idea by idea avoiding redundancy and unnecessary information. The first paragraph in particular is hard to read and it is very long (L30 to L84). See detailed comments below:

> Reply: We have shortened and modified the introduction as suggested, and hope the first paragraph is easier to read now.

L30. Please consider rewriting this sentence to increase the accuracy of the statement. For instance: Organic carbon buried in sediments underneath marine vegetation.

> Reply: Thank you for your suggestion. The sentence has been modified in order to improve the accuracy in the final manuscript. The first paragraph now reads: "Despite the fact that seagrass meadows cover only a 0.1% of the ocean surface, they are responsible of a 20% of the global carbon sequestration in marine sediments (Duarte et al., 2004; Kennedy et al., 2010) known as "blue carbon", which is defined as organic carbon buried in sediments underneath marine vegetation, like mangroves, saltmarshes and seagrass sediments (Duarte et al., 2004; Kennedy et al., 2010; Mcleod et al., 2011; Greiner et al., 2013). Carbon burial is the result of the combination of intense metabolic activity of the vegetation, high trapping capacity of allochthonous matter and an effective carbon preservation in sediments underneath meadows (Cebrian, 1999). Due to the enhanced deposition rates caused by the physical presence of the canopies in the water-column seagrass meadows capture suspended organic matter, which accumulates as organic matter in the sediment  (Romero et al., 1994; Pergent et al., 1997; Mateo et al.,

2006; Hendriks et al., 2008; Kennedy et al., 2010). Also the *in situ* production as plant growth due to primary production contributes to organic matter accumulation in the sediment (Greiner et al., 2013). There are species specific differences in carbon burial capacities and stock, for instance for P*osidonia oceanica* meadows a huge carbon storage capacity has been estimated, ranging from 40 to 770 kg $C_{org}$ $m^{-2}$, as the organic-rich soil accumulated beneath the canopy can be up to 6000 years old and reach a thickness of up to 13 metres (Mateo et al., 1997; Lo Iacono et al., 2008; Serrano et al., 2016)."

L33. remove dot before the references.

     Reply: Removed.

L34. add "an" before intense.

     Reply: Added to the text.

L34. Remove "together with excess production". I believe the authors meant high productivity rates, but the word excess is a subjective assessment that can lead to confusion

     Reply: Thank you for the suggestion, part of the sentence has been removed in the text.

L34. Remove "in seagrass meadows" because it is obvious

     Reply: Removed in the text.

L35. Increased compare to what? Consider replacing "increased" by "high"

     Reply: Thank you for the suggestion, "increased" changed by "high" in the text.

L35-L40. This statement is redundant with the one before ("high trapping capacity of allochthonous matter in seagrass meadows".

Reply: We have clarified the sentence removing the redundancy, see the revised first paragraph above.

L40. Consider removing: "elements such as"

Reply: Thank you for the suggestion, "elements such as" has been removed in the text.

L39. this last sentence hangs alone in the text and it is difficult to understand what it refers to. Please review: "together with in situ production due to their primary production (Greiner et al., 2013)."

Reply: Thank you for the remark, we have modified it in the text.

L43. The species names should always be in italics

Reply: We apologize for the format error. Format changed in the text.

L50. Unclear what it means "consistent estimates". Does it refer to methodology?

Reply: Indeed, we referred to methodologies. The statement has been modified in the text for clarity.

L56. Consider replacing "human processes" with "human activities".

Reply: Thank you for the suggestion. The recommended change has been added to the text.

L56. I believe this refers to the dynamics of the carbonate system but needs clarification.

Reply: Clarification added to the text.

L60. Two dots in a row, remove one.

Reply: We apologize for the format error. Dot removed in the text.

L85. Consider replacing "which are located in" to "from", as C. nodosa can also be found outside the Mediterranean.

Reply: Thank you for the suggestion. Change added to the text.

L96. Consider replacing "as ranging" to "to range"

Reply: Replacement added to the text.

L102. Add space between "Mediterranean meadows"

Reply: Space added to the text.

L124. Consider replacing "by the use of" to "using".

Reply: Change added to the text.

L132. Consider replacing "large" with "larger"

Reply: Term replaced in the text.

L136. Remove "the" before "two" as there are more seagrass species in the Mediterranean.

Reply: Thank you for the suggestion. "the" removed in the text.

L139 Remove "including the two species in the Mediterranean Sea".

Reply: Removed from the text.

**Methods**

General comment: In the abstract, it says that part of the data analysed in the study is its own data. But in the methods, it states that data is from published literature or published datasets. Does it mean the "own data" comes from previously published work? Is there any data collected in the field for the purpose of this study? All this needs clarification. Based on the information in the abstract I was hoping to see an assessment of how seagrass metabolism has changed through the years (authors have data since 1982) as a function of changes in the CO2 atmospheric concentrations "In this study we analyse the metabolism synthesized from published data on seagrass community metabolism and from own results to evaluate trends through time". If possible, it would be really interesting to include this.

Reply: We thank the reviewer for pointing this out. By "own data" the authors referred to all the data collected by the IMEDEA Global Change department (some of the data was published and some is unpublished). We have clarified the text, as the wording was confusing. We acknowledge that this might not be fully clear in the text as the wording was confusing and therefore we have clarified the text. This study brings in 3 unpublished data sets, 1 from Mallorca (W Med) but more importantly 2 from the Eastern basin, from Crete and Cyprus, and therefore expands the current knowledge of metabolic rates in the Eastern basin considerably. We did analyse the data for trends over time for changes in metabolism, but we did not find any significant results for the data collected with sensors. This could be due to the fact sensors are picking up a highly "composed" signal, as water column mixing makes it difficult to attribute measured metabolism to a single habitat. We did, however, find a difference over time (Year) for CR and GPP for the benthic chambers with the new analyses, but not NCP, which might indicate the changes are in opposite direction, leading to a similar NCP over time. As the dataset includes different methodologies, regions, highly variable sites and measurements done mostly in summer, it was difficult to get robust results for an unbalanced design, specifically evaluating the effect of season. Although theoretically there are seasonal trends, our results did not shown these trends due to the bias of the data set with more data available during summer compared to other seasons.

The paragraph now reads: "All data for benthic chamber deployments was extracted from the literature (published or submitted), while part of the sensor data for the metabolic parameters was extracted from the literature (published or submitted) while another part was obtained from unpublished data in the Western- but also more importantly Eastern Mediterranean Basin (Crete,

Cyprus; Table 1). Data available as oxygen concentration over time was processed and analysed to obtain the metabolic parameters, when this was not available we used reported values for metabolism."

L146. Site description: The way is written suggests that field data was specifically collected for this study (see comment above). If this is not the case, consider re-writing this part avoiding the use of terms like "sampling campaigns" or "sampling sites" and/or specifying that all this information comes from previous work. Furthermore, there is a high level of detail on the site description that (in my opinion) is unnecessary for a scientific paper. In case it is necessary for discussion, consider moving that info (such as the different status of protection of each site: SPA, Birds directive, ZEPA, LIC, ZEPIM, etc.) to the discussion section.

Reply: Thank you for the suggestions. The terms "sampling campaigns"/" sampling sites" have been replaced in the main text, except for the locations that were specifically collected for this study. We think the details of the site description are useful to have an environmental context about the sites where the data was specifically collected as this information is not available by referring to published literature. However, we agree the description of the other sites is too detailed and have re-arrange the section to highlight the information of the "new" sites and put them in the context of the type of existing sample locations.

This paragraph now reads: "We estimated metabolism from oxygen data of multiparametric sensors deployed in the Western and Eastern Mediterranean basin. In total we processed data from eight sites in Mallorca (Spain), two sites in Crete (Greece) and one in Cyprus (Republic of Cyprus).  All study sites were located in shallow sites, ranging from 2.9 metres depth (Punta Negra, Mallorca) to 15.7 metres depth (Cap Enderrocat, Mallorca). Multiparametric probes were measuring in either *Posidonia oceanica* and/or *Cymodocea nodosa* meadows (see Table 1).

Data was collected from published work, and collected during different periods ranging from 2011 to 2019 (for details see Table 1) and from dedicated sampling campaigns in 2016 in Mallorca (Western Mediterranean) and 2017 in the Eastern basin (Crete and Cyprus, see Table 1). The sampling site in Cyprus was located in Limassol, East Akrotiri bay, considered an impacted area affected by high anthropogenic pressures related to tourism and the construction of extensive coastal infrastructures. In Crete, Marathi and Kalami are considered as a single sampling site due to the proximity and similitude of the environmental factors of both sampling sites. This sampling station, located in Western Crete close (< 10 km) to the Port of Souda, is impacted by notably

sewage discharge, agriculture and industrial/chemical pollution; according to Simboura et al. (2016) this station is considered to have a moderate pressure index.  Maridati, the second station located in Crete is situated on the East side of the Island, in a pristine bay with no human coastal activity but affected by ensuing discharges of an ephemeral stream. The dedicated sampling campaign in Mallorca was in Cap Enderrocat, which forms part of an SPA (*Special Protection Area*) under the *Birds* Directive and is a SIC (site of Community Importance, Natura 2000), as well as Son Veri and Cala Blava, for which we extracted existing data, which are also protected and count with 11.5% of the total Posidonia meadows within the ZEPA Cap Enderrocat- Cap Blanc area. The other locations for which we extracted data ranged from pristine to impacted, Magalluf is in front of a touristic beach but the location of the sensors was sheltered behind an island (Isla Sa Porrassa).  Sant Elm is located in a relatively pristine area but near a sewage plant outlet. Pollença is in an enclosed bay affected by considerable organic input from the s´Albufereta wetlands, an emissary of the sewage plant, nearby harbour and urban area. Punta Negra is considered as a Natural Area of Special Interest (ANEI and a natural space protected by law by the Balearic Islands Government) while Sta. Maria, a bay located on the coast of Cabrera is the most pristine sampling area. Cabrera island is part of a Maritime and Terrestrial National Park located at the Cabrera Archipelago, and recognized internationally as ZEPA, LIC, Z.E.P.I.M (Special protection zones with importance for the Mediterranean and ZEC (Special zone of conservation). The sampling sites in the Mediterranean therefore include sites with different degrees of human impact and protected areas with very low anthropogenic impact."

L156. Add space after "Souda,"

    Reply: Space added in the text.

Fig 1. Add north arrow and latitude and longitude degrees in the axes. Missing reference for GEBCO 2020 in the reference section.

    Reply: North arrow and GEBCO2020 reference added. The final Fig. 1 with the longitude and latitude degrees added appeared too saturated, so the authors considered to keep the original map with the north arrow included

L188. Add "traits" after "habitat"

    Reply: Added to the text.

L183. Data analysis: Please add the accuracy (± SD) of the multiparametric sensors for each of the parameters used, especially for DO and pH. This is crucial for further interpretation.

Reply: We apologise for the lack of information in the text and included the accuracy of each sensor.

L187 - L189. Need to add methods for the habitat data.

Reply: we followed the procedure described in Hendriks et al. 2014 and added this information to the text.

Hendriks, I. E., Olsen, Y. S., Ramajo, L., Basso, L., Steckbauer, A., Moore, T. S., Duarte, C. M. (2014). Photosynthetic activity buffers ocean acidification in seagrass meadows. *Biogeosciences, 11*(2), 333-346. doi:10.5194/bg-11-333-2014

Table 1. Not sure what is the date format required by Biogeosciences but consider using MM/DD/YYYY.

Reply: Thank you for the concern, we have checked the date format required by Biogeosciences and it is DD/MM/YYYY. Therefore we have left the format as it is.

L211. Salinity is unitless. Remove units here and in Table 1

Reply: Thank you for the remark, salinity unites were removed from the text and in Table 1.

L223-L225. In the k and k660 calculations, what is the effect of the higher salinity found on each of the sites?

Reply: We appreciate your concern. In this study K and k600 calculations were chosen from the work published by Kihm and Körtzinger in 2010 and by Cole and Caraco in 1998, as they were the most suitable for coastal areas. These authors did not reflect on the effect of high salinities, specifying that the stronger dependence in the parameterizations is caused by elevated wind speeds, which is not our case. However, we truly believe this is an aspect that

should be included in future specific studies of the air-sea gas transference in high salinity areas.

L277. How were the 12 publications selected? Is this the total number of published works for P.oceanica and C.nodosa in the Mediterranean? If not, it will really help to include more data from seasons and regions understudied (for instance: studies with spring, fall, or winter data from the Eastern basin).

Reply: The 12 publications were selected after a thorough search and to the best of our knowledge they reflect the total number of published works with metabolic data for *P. oceanica* and *C. nodosa* in the Mediterranean Sea. We would greatly appreciate receiving information on additional studies if the reviewer noticed they´re not included at present.

This paragraph now reads: "We compiled data using the benthic chambers methodology from published literature, using publications in different states of progress from the group and through a search on the Web of Science and Google Scholar and found a total of 12 publications with data for *P. oceanica* and/or *C. nodosa* meadows. These studies were carried out from 2000 to 2019. Net Community Production (NCP) was generally estimated from changes in dissolved oxygen using the Winkler titration spectrophotometric method (Labasque et al., 2004). Benthic chambers enclose a section of the seagrass meadow, and flexible fitted plastic bags, not permeable for gases, assure the possibility of movement of the shoots inside, see details in the method section of each paper for the exact construction used.  The benthic chamber methodology has been more generally used to assess metabolism of seagrass meadows and the database of this study contains a total of 100 NCP estimations. We compare the data obtained between both methodologies. NCP, GPP and CR data were extracted from literature as well as accompanying biotic parameters."

L278. Add space before 12

Reply: Space added in the text.

L281. "In this work we add benthic chambers data to the body of literature," suggests that field data was collected, but no other explanation is given. See the comments above about clarifying this.

Reply: We agree this sentence was confusing, we added more details on the benthic chamber's methodology to the text. In fact, no unpublished data was used for the benthic chambers, only published literature, either from the IMEDEA Global Change group or outside.

L282-L285. I believe this sentence corresponds to a data analysis section, not to data compilation. Please add information on how the ANOVA assumptions were tested, especially the lack of independence from the time series data and data from the same site/season/region when comparing metabolic rates. Was any random factor considered? If not, the statistical analysis for the comparison of metabolic rates should be reviewed. For all statistical analyses done, please add information on how the residuals looked and if those met the assumptions of the correspondent analysis.

Reply: We moved the sentence to the Data analysis section. Furthermore, we revised all statistical analyses and used a more appropriate design as suggested. We used mixed models, through the lme4 package in R with random factors. For instance when we evaluate the difference between species for the sensor data we used "Sites" as a random factor as some sites had data for 1 species and some for both. We could not use mixed models with random factor for all the data due to unbalanced number of measurements and therefore used general linear models instead when not assigning random factors. We have added more information on the statistic outcome to the text as well (t values, degrees of freedom).

The phrase in the data analysis section now reads: "We used mixed models and general linear models with package lme4 in Rstudio to evaluate methodological, regional and species differences. We also analysed abiotic (wind and depth) and biotic parameters (shoot density and biomass) related to sensor data as there was more additional data associated to these measurements. As the data was not normally distributed according to the Shapiro-Wilk test, we log transformed data for GPP, NCP and CR before analysis."

L284. Are density and shoots the same measurement? How were all these parameters measured? See the comment above about the need to add methods for the habitat data.

Reply: Thank you for the remark, the notation has been erroneous and, in fact, it should be "shoot density". We clarified the text. As all the benthic chamber data comes from published data, we have extracted the details for biotic parameters from the papers as well.

Table 2. Two decimals are enough for temperature, salinity, and depth. Also, remove units in salinity. Consider adding here or in the text the characteristics of the chambers (i.e. flexibility and material).

Reply: Thank you for the remark. Superfluous decimals and salinity units were removed from the text. As the benthic chamber data is published, we have added some sentences on the general construction of benthic chambers and referred for specifics to the respective papers.

**Results:**

General comment: There are methods written in the Results section. It would be better to move that to the methods section. I have serious doubts about the use of non-significant results in one-way ANOVAs to pooling datasets in data that is (for what I can see in the methods section) not independent. The results on habitat traits and abiotic parameters used (pH for instance) and many of the logistic regressions (temperature, shoot density, etc.) are missing and should be added. Finally, I would suggest, in order to gain clarity, to summarize section 3.1 in a Table and keep consistency on the use of written numbers.

Reply: we agree summarizing section 3.1 improves the readability of the paper and have moved that information to Table A2 in the appendixes. We apologize for not including the linear regression mentioned in the earlier version but we believe that as this information was not significant, it was therefore without relevance for the paper and would only add confusion. We have now added all statistical data, even when not significant.

The results section with the revised statistics for the sensor part now reads: "As sensor data were collected in the water column, with lateral movement between habitats of water masses, and there were no significant differences, GPP ($\chi^2$=0.11, $p$=0.74), CR ($\chi^2$=0.50, $p$=0.48) and NCP ($\chi^2$=0.06, $p$=0.81), for any of the three metabolic parameters between the two species (*P. oceanica* and *C. nodosa*), tested with "Site" as random factor, we didn´t divide the sensor data for the two species. Gross Primary Productivity ($\chi^2$=1.59, $p$=0.21), NCP ($\chi^2$=0.13, $p$=0.71) and respiration (CR; $\chi^2$=0.15, $p$=0.70) were similar between the Eastern and Western Mediterranean basins (Fig. 3), probably due to the high variability between sites (used as random factor), and skewed distribution of seasonal data between the regions. Although when only data for summer where tested

this similarity persisted. The highest GPP rates (Mean ± SD) occurred during spring with 453.92 ± 233.3 mmol $O_2$ m$^{-2}$ day$^{-1}$ and in fall with 241.1 ± 156.4 mmol $O_2$ m$^{-2}$ day$^{-1}$, the corresponding CR rates for spring and fall were 61.5±379 mmol $O_2$ m$^{-2}$ day$^{-1}$ and 180.4 mmol $O_2$ m$^{-2}$ day$^{-1}$ respectively. Productivity was higher than respiration for all the seasons reflected in positive averaged NCP rates and confirming that seagrass meadows tend to be autotrophic ecosystems, with the highest values found during spring and summer with 408.08 ± 454.9 mmol $O_2$ m$^{-2}$ day$^{-1}$ and 225.2 ± 280.9 mmol $O_2$ m$^{-2}$ day$^{-1}$, respectively. However, due to the high variability, NCP ($\chi^2$=0.27, $p$=0.97), CR ($\chi^2$=0.61, $p$=0.89) and GPP ($\chi^2$=5.45, $p$=0.24) were not different between seasons while the mean P/R ratio was above 1 (1.3 ± 9.7), confirming the tendency of net autotrophy. Additionally, no significant trends were found for any of the metabolic parameters measured during summer (to prevent the influence of seasonal fluctuations) over time using measurement year as continuous variable and "Site" as random factor, with NCP ($\chi^2$=0.57, $p$=0.45), CR ($\chi^2$=0.49, $p$=0.48) and GPP ($\chi^2$=2.46, $p$=0.12). Maximum GPP in the Western basin in summer was 483.10 ± 705.3 mmol $O_2$ m$^{-2}$ day$^{-1}$, while very variable and not significantly different from the Eastern basin, with averages more than two times higher (175.74±110.3 mmol $O_2$ m$^{-2}$ day$^{-1}$).  NCP in the Eastern basin was 349.45 ± 393.9 mmol $O_2$ m$^{-2}$ day$^{-1}$ and in the Western basin (225.2± 280.9 mmol $O_2$ m$^{-2}$ day$^{-1}$). The high standard error values reflect the high variability found in the individual studies. During summer, NCP in the Eastern basin ranged from -293.7 mmol $O_2$ m$^{-2}$ day$^{-1}$ to 713.6 mmol $O_2$ m$^{-2}$ day$^{-1}$ and fluctuated from 23.5 to 1207.4 mmol O2 m$^{-2}$ day$^{-1}$ in the Western basin. In the Eastern Mediterranean basin, only data recorded in summer was available, with an NCP rate of 349.45 ± 393.9 $O_2$ m$^{-2}$ day$^{-1}$; the GPP rate 175.74±110.3 mmol $O_2$ m$^{-2}$ day$^{-1}$ was lower than CR 173.7±431.6 mmol $O_2$ m$^{-2}$ day$^{-1}$, indicating that these seagrass communities tend to be net autotrophic during this period, reflected in an average P/R ratio just above 1 (1.01±0.4). The temperature recorded during the highest NCP measurement in the Western basin was 26.6ºC, which is close, even though a bit higher, to the optimal value reported for *P. oceanica* of 25.8 ºC  (Savva et al., 2018). For the Eastern Mediterranean basin, the highest GPP obtained was 357.31 mmol $O_2$ m$^{-2}$ day$^{-1}$ at Limassol station (Cyprus) during September and the *in situ* temperature registered at that moment was 27.7ºC, which was not the highest temperature registered in the Eastern basin (28.5ºC) but higher than the mean temperature in the Eastern basin during the summer sampling campaign (25.9±0.8 ºC). The lowest GPP values found in the Western and Eastern regions were different, we found a negative GPP of 3.81 mmol $O_2$ m$^{-2}$ day$^{-1}$  for the Western basin in the Cala Blava station (Mallorca) during spring whereas the lowest GPP value in the Eastern basin was 14.12 mmol $O_2$ m$^{-2}$ day$^{-1}$  in Marathi station (Crete) in summer;  temperatures during both measurements were similar with less than one Celsius degree of difference between them (26.7ºC in Marathi station (Crete) and 25.9ºC in Pollença station (Mallorca). We tested with individual regression models for the effect of

temperature, which did not significantly affect GPP ($t_{df=67}$=-0.035, $p$=0.97), and NCP ($t_{df=64}$=1.86, $p$=0.07) but did affect CR ($t_{df=64}$=2.29, $p$>0.05) and had a significant effect on NCP ($t_{df=44}$=3.59, $p$<0.001) when only the data for summer was included (See Appendices, Fig. A5). Depth affected GPP ($t_{df=63}$=4.36, $p$<0.001, Figure A3) but not NCP ($t_{df=64}$=1.09, $p$=0.28) or CR ($t_{df=64}$=-1.81, $p$=0.08). Windspeed did not drive metabolic rates with $t_{df=63}$=-0.69, $p$=0.49; $t_{df=64}$=-1.05, $p$=0.30 and $t_{df=64}$=-0.59, $p$=0.56 respectively for GPP, NCP and CR. Shoot density and biomass are correlated and neither variable was related with metabolic rates, with p values between 0.60 and 0.83."

And for the benthic chamber part: "We found significant differences for NCP ($t_{df=98}$=3.85, $p$<0.001) and GPP ($t_{df=65}$=3.50, $p$<0.001; Fig. 4) between *P. oceanica* and *C. nodosa* productivity, but not for respiration ($t_{df=65}$=-0.05, $p$=0.96). As we did not have *C. nodosa* data for the Eastern Mediterranean basin we only examined *P. oceanica* to distil patterns between Eastern and Western Mediterranean basin regions. There were no significant differences for NCP ($\chi^2$=0.15, $p$=0.70), GPP ($\chi^2$=0.20, $p$=0.65) and CR ($\chi^2$=1.99, $p$=0.16) for *Posidonia* incubations between Eastern and Western regions (Fig. 5), due to the high variability between sites, which was incorporated in the model as a random factor. At a seasonal scale, there were no significant differences for NCP, GPP or CR for *C. nodosa* with NCP ($\chi^2$=0.22, $p$=0.90), GPP ($\chi^2$=0.49, $p$=0.78) and CR ($\chi^2$=0.16, $p$=0.93). Production was lower than respiration during fall and spring, this was reflected in the averaged NCP, with a negative rate (-9.2 ±23.0 mmol $O_2$ m$^{-2}$ day$^{-1}$), revealing that the *C. nodosa* community tends to be net heterotrophic, also reflected in the average P/R ratio below 1 (-1.05±1.8). There were no significant differences between NCP ($\chi^2$=3.95, $p$=0.41) and CR ($\chi^2$=6.91, $p$=0.14) in different seasons (with year as random factor) for *P. oceanica*, however GPP ($\chi^2$=12.11, $p$<0.05) was different (Figure A6). For the Western basin, averaged NCP was (19.6 ±28.2 mmol $O_2$ m$^{-2}$ day$^{-1}$). The average GPP (66.6 ±28.2 mmol $O_2$ m$^{-2}$ day$^{-1}$) was higher than the CR rate (-13.9±57.4 mmol $O_2$ m$^{-2}$ day$^{-1}$) which reflect the tendency of *P. oceanica* communities to be net autotrophic. There were no statistical differences between monthly rates of NCP ($t_{df=65}$=-1.59, $p$=0.12), CR ($t_{df=33}$=1.16, $p$=0.26) and GPP ($t_{df=33}$=-0.30, $p$=0.76) for *P. oceanica* (Figure A6). Similarly to the sensor data, temperature was not correlated with productivity  NCP ($t_{df=21}$=-1.14, $p$=0.27), and GPP ($t_{df=19}$=-1.00, $p$=0.33), but was affecting CR ($t_{df=19}$=-2.66, $p$<0.05). For chamber incubations we found an evolution over time using year (See Appendices Fig. A4, A5) with GPP ($t_{df=45}$=-4.99, $p$<0.001) and CR ($t_{df=45}$=2.54, $p$<0.05) but not NCP ($t_{df=78}$=0.17, $p$=0.86). As more data was available for *P. oceanica*, we were able to analyse its metabolic rates regionally (Eastern and Western Mediterranean basins) and temporally (seasonally, monthly, and yearly). For the Eastern basin, we found the highest *P. oceanica* individual NCP value during spring with a metabolic rate of 63.85 mmol $O_2$ m$^{-2}$ day$^{-1}$ and the lowest was found during fall with 27.04 mmol $O_2$ m$^{-2}$ day$^{-1}$. For CR, during summer the highest value was -25.55 mmol

O$_2$ m$^{-2}$ day$^{-1}$ and the lowest was -106.64 mmol O$_2$ m$^{-2}$ day$^{-1}$ during summer. In the Western region, where the higher amount of data was available, we found a maximum NCP for *P. oceanica* of 136.85 mmol O$_2$ m$^{-2}$ day$^{-1}$ during summer 2001 and a minimum value of -15.4 mmol O$_2$ m$^{-2}$ day$^{-1}$ during the same summer. For the CR in this region for Posidonia, we found values ranging from -141.9 mmol O$_2$ m$^{-2}$ day$^{-1}$ in summer to 150.8 mmol O$_2$ m$^{-2}$ day$^{-1}$ in fall."

We have also corrected the abstract and discussion to reflect for instance the fact that, due to using "Site" as random factor in the region analysis, with the high variability, there are no regional differences observed for metabolic rates.

L295-L298. All this info can be removed or moved to the Methods section. If the data is available, please add the correspondent link.

   Reply: Information removed and summarized in the table A2, in the appendixes. The final database will be available through the repository with the correspondent link upon acceptance of the paper.

L310. In the stats analysis, please provide more details: degrees of freedom, F-values, Sum or Mean of Squares for ANOVA, etc. This information can go in a Table into supplementary materials.

   Reply: We have revised the statistics, and updated the results section including t-values (for linear regression models) and $\chi^2$-values (for mixed models) with accompanying degrees of freedom. We think providing an additional table in the appendix might be confusing as there are many analyses and thus there would have to be several tables or composed tables. We are willing to include these though if the reviewer thinks this would improve the clarity of the paper.

L310. See my general comment above about merging datasets based on simple one-way ANOVAs.

   Reply: We agree simple ANOVAs are maybe not the best way to analyse the data. However, we do think that merging some data, is justified. For instance, in the case of the sensor data from the two species. Even though the underlying idea was to capture species-specific metabolic rates, In practice this has proven to be extremely difficult due to lateral movement of water masses. Even in large sandy areas in Posidonia meadows the metabolic signal of the meadow is noticeable (data not shown, personal experience of the authors) and it is difficult to separate the components (species specific productivity) contributing to the ecosystem productivity measured in the water column. So, in this case merging this data has a biologically sound reason, backed up by the

statistical test. We did, however, revise the statistics as we do agree the previous analyses were too simple. Nested ANOVAs or ANCOVAs as well as mixed models are far more appropriate. We have decided to use mixed models to be able to include random factors.

L321. See my general comment above about merging datasets based on simple one-way ANOVAs.

   Reply: see comments above

L328. Replace "didn´t" by "did not".

   Reply: Replaced in the text.

L330-L333. If possible, I would suggest moving the methods and results related to temperature from the appendix to the main manuscript. The finding of temperature not affecting metabolic parameters in the Western basin is very relevant to the work done and is very interesting.

   Reply: We agree with the reviewer that the lack of correlation between metabolic parameters and temperature is interesting and definitely unexpected for us. However, we fear this is due to the unbalance of the data over the seasons, with a range of temperatures within different seasons and their corresponding biological activities of the seagrass. We would therefore prefer to leave this graph in the appendix.

L329. Remove capital letter from "Addition"

   Reply: Capital letter removed in the text.

L331. Replace "none" with "any"

   Reply: Replaced in the text.

L346. I would suggest removing "and act as carbon sinks" as this was not studied.

   Reply: Removed from the text.

L365. Replace "didn´t" by "did not"

Reply: Replaced in the text.

L369. "Except for the summer" hangs alone and it is difficult to know what it means.

Reply: Thank you for the remark, removed in the text.

L373. See my general comment above about merging datasets based on simple one-way ANOVAs.

Reply: as commented above

L375. Keep consistency on the number of decimals used for each parameter.

Reply: Corrected in the text.

**Discussion**

General comment: There are results (I believe from the logistic regressions) written in the Discussion section that should be moved to the Results. Also, it would help the readers to have a first paragraph on the discussion with the take-home message.

Reply: We appreciate your comment. Some of the results presented in the discussion section have been added in the results section. In addition, a first paragraph in the discussion with the take-home message have been added.

L413. Replace "didn´t" by "did not"

Reply: Replaced in the text.

L417. This statement about the 10m distance among seagrass meadows is very confusing. From Table 1, only two sites presented both species. Please clarify what do you mean here.

Reply: Thank you for the remark, clarification added in the text.

L423. Replace "didn´t" by "did not"

Reply: Replaced in the text.

L430. I would suggest removing "and act as carbon sinks".

Reply: Removed in the text.

L432. Keep consistency in the use of acronyms.

Reply: We appreciate your comment and revised all the acronyms in the text.

L439. These results are not presented anywhere.

Reply: Thank you for pointing this out. Results have been included in the Results section.

L440 – L447. These results need to be presented in the Results section

Reply: We appreciate your comment. These results have been included in the Results section.

L441. Add space after comma, and remove dot before comma

Reply: Thank you for the remark, corrected in the text.

L446. The results of the biotic parameters related to metabolism are really surprising and it would be interesting to discuss them further.

Reply: As mentioned in the text, biotic parameters like shoot density and biomass were not determinant for GPP, CR nor NCP (p>0.1), which underlines the effect of lateral advection and mixing of water masses influencing the net signal measured by the multiparametric probes. However, we appreciate your comment and believe that this could be included in further studies with more available biotic data in order to see if there is more dependence.

L454 Replace "wasn´t" by "was not"

Reply: Replaced in the text.

L455. See my comment in methods about the bibliographic research. Does this mean that no benthic chambers have ever been used in *C. nodosa* in the

Eastern basin? If this is the case, the results presented in this work are even more important and this should be highlighted as one of the outcomes.

Reply: We found data on benthic chambers for *Posidonia oceanica* used in the Eastern basin in the publication by Apostolaki et al., 2010, we included the reference below. On the other hand, we did not find published data in the Eastern basin with sensors neither for *P.oceanica* or *C.nodosa.*

Apostolaki, E. T., Holmer, M., Marbà, N., & Karakassis, I. (2010). Metabolic imbalance in coastal vegetated (*Posidonia oceanica*) and unvegetated benthic ecosystems. *Ecosystems, 13*(3), 459-471.

L458. Replace dot by comma

Reply: Replaced in the text.

L459. Avoid repeating results in the discussion section.

Reply: Removed in the text.

L471. Please cite the correspondent literature.

Reply: Clarified in the text.

L486. Remove dot after column

Reply: Dot removed from the text.

L515. Replace "didn´t" by "did not"

Reply: Replaced in the text.

L518. Remove "a" before "more".

Reply: Removed from the text.

L544. Remove "prevention"

Reply: Removed from the text.

**Appendices**

Appendix B is really scattered and the results of the higher GPP with depth seem to be driven by only 1 depth (15m). Is this only driven by one site?

Reply: Thank you for the remark. We agree there is a high variability in the data. To clarify, the GPP values at 15m depth are measured at the same site for 11 consecutive days and we consider them relatively robust. The significant relationship of GPP with depth holds for the new statistical analysis, which is why we have decided to keep the figure.

Appendix D. remove capital letter from oceanica.

Reply: Thank you for the remark, format changed in the text.

Apostolaki, E. T., Holmer, M., Marbà, N., & Karakassis, I. (2010). Metabolic imbalance in coastal vegetated (*Posidonia oceanica)* and unvegetated benthic ecosystems. *Ecosystems, 13*(3), 459-471.

---

## Author Response (AR1)

**Reply to Comment on bg-2021-60**

**Referee #1**

The manuscript by Escolano-Moltó et al. presents a synthesis of seagrass metabolic data from previously published work and/or datasets in the Mediterranean relative to two seagrass species (*Posidonia oceanica* and *Cymodocea nodosa*) using two methodologies (benthic chambers and multiparametric sensors). This is a very relevant topic in the current context of climate change in relation to carbon sequestration in coastal areas, and the work presented has a considerable amount of data and results that fit within the scope of Biogeosciences. While the seagrass metabolic data is not particularly novel, the comparisons among methods, species, and regions (Mediterranean basin) are very important. However, there is a major flaw in the statistical approach used and how this is used to pooling datasets. As presented in the manuscript, the ANOVA analysis is not considering the lack of independence in the data from the same season, site, or region and should be reviewed. Depth should also be considered as a covariate as it is most likely related to the metabolic rates due to the light availability. Increasing the accuracy in the statistics presented is essential for the interpretation of the results presented here, especially because datasets are pooled based on those analyses and then further analyses are done. Therefore, the results presented are built over potentially incorrect statistical analysis, and, right now, it is not possible to evaluate the accuracy of the entire set of results presented. If ANOVA assumptions cannot be met, consider using a different statistical approach (e.g. mixed models) and present the results accordingly. Especially critical is the pooling of datasets, if possible, this should be avoided and instead, grouping factors or separate analysis should be considered. Additionally, the main text structure needs revision (see specific comments below). In particular, there is a lot of information on the methods section that is missing in the Results (e.g. habitat traits measured, logistic regressions between abiotic and biotic parameters, pH data). Also, there are Results (including stats) presented in the Discussion section. Throughout the text, there are several typos and constant misuse of species names, which appear sometimes complete and others shortened, and many times italics are not used. I believe the work presents interesting data, and so, the analyses could be revised to improve the way results are presented and discussed in the manuscript. Hopefully, my suggestions help to improve the manuscript. All my comments are made with this purpose.

Before detailing our replies to the reviewer I would like to indicate that through the revision of our data for the revised manuscript we have made the hard decision to exclude sensor data with positive oxygen signals during the night time. As we already mentioned in methods, results and discussion sections in the last version, the sensor data has the disadvantage of picking up oxygen

concentrations from water volumes drifting past by lateral advection. A positive signal during the night time is a clear indication of this problem and thus we have excluded data where we suspected a big influence of lateral advection. Currents are not usually intense in the Mediterranean and in our opinion this is not a common problem with the dataset and does not invalidate sensor results. However, we have wanted to be on the cautious side and only present data we absolutely confide in. Therefore the database has decreased somewhat in size and this has also meant all statistics and figures have been re-done and some results have changed. We apologize for this, and also the delay it has caused in our revision. The extensive changes made in the manuscript have also caused us to reconsider the order of authors as you may have noticed. However we think the extensive reworking of the manuscript has vastly improved the quality of the analyses and the conclusions are much more robust.

Reply: We thank the referee for the helpful comments, we have restructured the text as suggested, and taken all the specific comments into account. We understand the concerns about the ANOVA analyses and have redone the analyses using mixed models (package lme in R) and included depth as a factor. See replies to the specific comments below.

**Abstract:**

L14. I would recommend replacing ": "Through their metabolic activity, they …" with "Seagrasses". As it is written now, the statement neglects the fact that carbon stored in sediments can come from external sources and that the buffer of low pH can also occur due to other processes not related to the seagrass aerobic metabolism.

Reply: Thank you for the suggestion, we have modified the text accordingly. The sentence now reads: "Seagrasses can act as carbon sinks; buffer lowering pH values during the day and store carbon in the sediment underneath their meadows."

L15. This is a long sentence that could be re-written to increase clarity. For instance: In this study, we analysed published and own (unpublished?) data on seagrass community metabolism to evaluate trends through time of these two species comparing two methodologies: benthic chambers and multiparametric sensors.

Reply: Thank you for the suggestion. The modification has been included in the manuscript. The sentence now reads: "In this study, we analysed published and previously unpublished own data on seagrass community

metabolism to evaluate trends through time of these two species comparing two methodologies: benthic chambers and multiparametric sensors."

L19. remove "with no significant results despite the clear visual trends."

Reply: Modified in the text.

L21. Add a comma before whereas

Reply: Added to the text.

L23. add "the" before highest or replace by higher

Reply: Added to the text.

L23 - L24. write the complete species name in italics and remove the genus (i.e. P.oceanica, C. nodosa)

Reply: This was modified in the text.

**Introduction**

General comment: The introduction is long, there is a lot of information and it is difficult to follow the flow of ideas. This is especially the case around the importance of seagrass aerobic metabolism related to (1) carbon burial in sediments and (2) buffering of low pH. Both processes are related to primary productivity, however, there are differences among them that right now are unclear in the text. I would recommend reviewing the text, try to shorten it, and present idea by idea avoiding redundancy and unnecessary information. The first paragraph in particular is hard to read and it is very long (L30 to L84). See detailed comments below:

Reply: We have shortened and modified the introduction as suggested, and hope the first paragraph is easier to read now.

L30. Please consider rewriting this sentence to increase the accuracy of the statement. For instance: Organic carbon buried in sediments underneath marine vegetation.

Reply: Thank you for your suggestion. The sentence has been modified in order to improve the accuracy in the final manuscript. The first sentences of the first paragraph now read: "A fifth of the global carbon sequestration in marine sediments (Duarte et al., 2004; Kennedy et al., 2010) can be attributed to seagrass meadows, despite the fact that they cover only a 0.1% of the ocean surface. This "blue carbon", which is defined as organic carbon buried in sediments underneath marine vegetation (Duarte et al., 2004; Kennedy et al., 2010; Mcleod et al., 2011; Greiner et al., 2013) is the result of the combination of intense metabolic activity of the vegetation, high trapping capacity of allochthonous matter and an effective carbon preservation in sediments underneath meadows (Cebrian, 1999).."

L33. remove dot before the references.

Reply: Removed.

L34. add "an" before intense.

Reply: Added to the text.

L34. Remove "together with excess production". I believe the authors meant high productivity rates, but the word excess is a subjective assessment that can lead to confusion

Reply: Thank you for the suggestion, part of the sentence has been removed in the text.

L34. Remove "in seagrass meadows" because it is obvious

Reply: Removed in the text.

L35. Increased compare to what? Consider replacing "increased" by "high"

    Reply: Thank you for the suggestion, "increased" changed by "high" in the text.

L35-L40. This statement is redundant with the one before ("high trapping capacity of allochthonous matter in seagrass meadows".

    Reply: We have clarified the sentence removing the redundancy, see the revised first paragraph above.

L40. Consider removing: "elements such as"

    Reply: Thank you for the suggestion, "elements such as" has been removed in the text.

L39. this last sentence hangs alone in the text and it is difficult to understand what it refers to. Please review: "together with in situ production due to their primary production (Greiner et al., 2013)."

    Reply: Thank you for the remark, we have modified it in the text.

L43. The species names should always be in italics

    Reply: We apologize for the format error. Format changed in the text.

L50. Unclear what it means "consistent estimates". Does it refer to methodology?

    Reply: Indeed, we referred to methodologies. The statement has been modified in the text for clarity.

L56. Consider replacing "human processes" with "human activities".

Reply: Thank you for the suggestion. The recommended change has been added to the text.

L56. I believe this refers to the dynamics of the carbonate system but needs clarification.

Reply: Clarification added to the text.

L60. Two dots in a row, remove one.

Reply: We apologize for the format error. Dot removed in the text.

L85. Consider replacing "which are located in" to "from", as C. nodosa can also be found outside the Mediterranean.

Reply: Thank you for the suggestion. Change added to the text.

L96. Consider replacing "as ranging" to "to range"

Reply: Replacement added to the text.

L102. Add space between "Mediterranean meadows"

Reply: Space added to the text.

L124. Consider replacing "by the use of" to "using".

Reply: Change added to the text.

L132. Consider replacing "large" with "larger"

Reply: Word replaced in the text.

L136. Remove "the" before "two" as there are more seagrass species in the Mediterranean.

Reply: Thank you for the suggestion. "the" removed in the text.

L139 Remove "including the two species in the Mediterranean Sea".

Reply: Removed from the text.

**Methods**

General comment: In the abstract, it says that part of the data analysed in the study is its own data. But in the methods, it states that data is from published literature or published datasets. Does it mean the "own data" comes from previously published work? Is there any data collected in the field for the purpose of this study? All this needs clarification. Based on the information in the abstract I was hoping to see an assessment of how seagrass metabolism has changed through the years (authors have data since 1982) as a function of changes in the CO2 atmospheric concentrations "In this study we analyse the metabolism synthesized from published data on seagrass community metabolism and from own results to evaluate trends through time". If possible, it would be really interesting to include this.

Reply: We thank the reviewer for pointing this out. By "own data" the authors referred to all the data collected by the IMEDEA Global Change department (some of the data was published and some is unpublished). We have clarified the text, as the wording was confusing. We acknowledge that this might not have been fully clear in the text as the wording was confusing. This study brings in 3 unpublished data sets, 1 from Mallorca (W Med) but more importantly 2 from the Eastern basin, from Crete and Cyprus, and therefore expands the current knowledge of metabolic rates in the Eastern basin considerably. We did analyse the data for trends over time for changes in metabolism, but we did not find any significant results for the productivity (GPP, NCP) data collected with sensors. This could be due to the fact sensors are picking up a highly "composed" signal, as water column mixing makes it difficult to attribute measured metabolism to a single habitat. We did, however, find a difference over time (Year) for respiration (CR) for sensor data, with increasing values over the years (Figure A6). We also found a trend (not significant) over time for CR and GPP for the benthic chambers with the new

analyses, but not NCP, both CR as well as GPP decreasing, which is in contrast with the sensor data for CR and in contrast with expectations. As the dataset includes different methodologies, regions, highly variable sites and measurements done mostly in summer, it was difficult to get robust results for an unbalanced design, specifically evaluating the effect of season. Although theoretically there are seasonal trends, our results did not shown these trends due to the bias of the data set with more data available during summer compared to other seasons.

The paragraph now reads: "All data for benthic chamber deployments was extracted from the literature (published or submitted), while part of the sensor data for the metabolic parameters was extracted from the literature (published or submitted) while another part was obtained from unpublished data in the Western- but also more importantly Eastern Mediterranean Basin (Crete, Cyprus; Table 1). Data available as oxygen concentration over time was processed and analysed to obtain the metabolic parameters, when this was not available we used reported values for metabolism."

L146. Site description: The way is written suggests that field data was specifically collected for this study (see comment above). If this is not the case, consider re-writing this part avoiding the use of terms like "sampling campaigns" or "sampling sites" and/or specifying that all this information comes from previous work. Furthermore, there is a high level of detail on the site description that (in my opinion) is unnecessary for a scientific paper. In case it is necessary for discussion, consider moving that info (such as the different status of protection of each site: SPA, Birds directive, ZEPA, LIC, ZEPIM, etc.) to the discussion section.

Reply: Thank you for the suggestions. The terms "sampling campaigns"/" sampling sites" have been replaced in the main text, except for the locations that were specifically collected for this study. We think the details of the site description are useful to have an environmental context about the sites where the data was specifically collected as this information is not available by referring to published literature. However, we agree the description of the other sites is too detailed and have re-arrange the section to highlight the information of the "new" sites and put them in the context of the type of existing sample locations.

As the manuscript text has changed so substantially we do not highlight a specific paragraph in this reply, but refer to the new methodological section.

L156. Add space after "Souda,"

Reply: Space added in the text.

Fig 1. Add north arrow and latitude and longitude degrees in the axes. Missing reference for GEBCO 2020 in the reference section.

Reply: We have completely changed Figure 1 and made a new figure in Matlab with the location of the sample sites and latitude, longitude, north arrow and depth isobars.

L188. Add "traits" after "habitat"

Reply: Added to the text.

L183. Data analysis: Please add the accuracy (± SD) of the multiparametric sensors for each of the parameters used, especially for DO and pH. This is crucial for further interpretation.

Reply: We apologise for the lack of information in the text and included the accuracy of each sensor.

L187 - L189. Need to add methods for the habitat data.

Reply: we followed the procedure described in Hendriks et al. 2014 and added this information to the text.

Hendriks, I. E., Olsen, Y. S., Ramajo, L., Basso, L., Steckbauer, A., Moore, T. S., Duarte, C. M. (2014). Photosynthetic activity buffers ocean acidification in seagrass meadows. *Biogeosciences, 11*(2), 333-346. doi:10.5194/bg-11-333-2014

Table 1. Not sure what is the date format required by Biogeosciences but consider using MM/DD/YYYY.

Reply: Thank you for the concern, we have checked the date format required by Biogeosciences and it is DD/MM/YYYY. Therefore we have left the format as it is.

L211. Salinity is unitless. Remove units here and in Table 1

Reply: Thank you for the remark, salinity unites were removed from the text and in Table 1.

L223-L225. In the k and k660 calculations, what is the effect of the higher salinity found on each of the sites?

Reply: We appreciate your concern. In this study K and k600 calculations were chosen from the work published by Kihm and Körtzinger in 2010 and by Cole and Caraco in 1998, as they were the most suitable for coastal areas. These authors did not reflect on the effect of high salinities, specifying that the stronger dependence in the parameterizations is caused by elevated wind speeds, which is not our case. However, we do believe this is an aspect that should be included in future specific studies of the air-sea gas transference in high salinity areas.

L277. How were the 12 publications selected? Is this the total number of published works for P.oceanica and C.nodosa in the Mediterranean? If not, it will really help to include more data from seasons and regions understudied (for instance: studies with spring, fall, or winter data from the Eastern basin).

Reply: The 12 publications were selected after a thorough search and to the best of our knowledge they reflect the total number of published works with metabolic data for *P. oceanica* and *C. nodosa* in the Mediterranean Sea. We would greatly appreciate receiving information on additional studies if the reviewer noticed they´re not included at present.

This paragraph now reads: "Data for the metabolic parameters was extracted from the literature, through a literature search on SCOPUS and the Web of Science using the keywords "Posidonia", OR "Cymodocea", OR "Seagrass", AND "Productivity", OR "Metabolism" and manually screened for data on metabolism in the Mediterranean basin. This database was extended with submitted data and data from dedicated sampling campaigns in 2016 in Mallorca (Western Mediterranean) and 2017 in the Eastern basin (Crete and Cyprus, see Table 1, Fig. 1). We also compiled data from multiparametric sensors collected during different periods ranging from 2011 to 2019 (for details see Table 1). While data using the benthic chambers methodology had a higher number of literature studies, with a total of 12 publications with data for *P. oceanica* and/or *C. nodosa* meadows (for details see Table 2), and a wider temporal cover with studies carried out from 1982 to 2019. Importantly, this study adds new data on Mediterranean seagrasses metabolism in the Eastern Mediterranean Basin (Crete, Cyprus; Table 1), where little data has been

published before. Data available as oxygen concentration over time was processed and analysed to obtain the metabolic parameters, when this was not available, we used the reported metabolic rates."

L278. Add space before 12

      Reply: Space added in the text.

L281. "In this work we add benthic chambers data to the body of literature," suggests that field data was collected, but no other explanation is given. See the comments above about clarifying this.

      Reply: We agree this sentence was confusing, we added more details on the benthic chamber's methodology to the text. In fact, no unpublished data was used for the benthic chambers, only published literature, either from the IMEDEA Global Change group or outside.

L282-L285. I believe this sentence corresponds to a data analysis section, not to data compilation. Please add information on how the ANOVA assumptions were tested, especially the lack of independence from the time series data and data from the same site/season/region when comparing metabolic rates. Was any random factor considered? If not, the statistical analysis for the comparison of metabolic rates should be reviewed. For all statistical analyses done, please add information on how the residuals looked and if those met the assumptions of the correspondent analysis.

      Reply: We moved the sentence to the Data analysis section. Furthermore, we revised all statistical analyses and used a more appropriate design as suggested. We used mixed models, through the lme4 package in R with random factors. For instance when we evaluate the difference between species for the sensor data we used "Sites" as a random factor as some sites had data for 1 species and some for both. We could not use mixed models with random factor for all the data due to unbalanced number of measurements and therefore used general linear models instead when not assigning random factors. We have added more information on the statistic outcome to the text as well (t values, degrees of freedom).

      The phrase in the data analysis section now reads: "We used mixed linear models with package lme4 in the R environment (R core team, 2021) to evaluate differences between methods, regions and species. To reflect the variability between study approaches and sampling procedures and therefore

variability in the precision of outcome of each study, we used a linear model where publication was included as random effect unless specified differently. We also analysed abiotic (wind, pH, depth) parameters related to sensor data as there was more additional data associated to these measurements. As the data was not normally distributed according to the Shapiro-Wilk test, we log transformed data for GPP, and CR before analysis. NCP could not be log transformed due to negative values."

L284. Are density and shoots the same measurement? How were all these parameters measured? See the comment above about the need to add methods for the habitat data.

Reply: Thank you for the remark, the notation has been erroneous and, in fact, it should have been "shoot density". As all benthic chamber data comes from published data, the details for biotic parameters were extracted from the papers as well. In this revised manuscript we have left out this analysis on the base that we have too little data to evaluate the effect of this parameter properly. Furthermore, the manuscript contains a lot of information already and including this analysis does not add enough new information while it distracts from the main message.

Table 2. Two decimals are enough for temperature, salinity, and depth. Also, remove units in salinity. Consider adding here or in the text the characteristics of the chambers (i.e. flexibility and material).

Reply: Thank you for the remark. Superfluous decimals and salinity units were removed from the text. As the benthic chamber data is published, we have added some sentences on the general construction of benthic chambers and referred for specifics to the respective papers.

**Results:**

General comment: There are methods written in the Results section. It would be better to move that to the methods section. I have serious doubts about the use of non-significant results in one-way ANOVAs to pooling datasets in data that is (for what I can see in the methods section) not independent. The results on habitat traits and abiotic parameters used (pH for instance) and many of the logistic regressions (temperature, shoot density, etc.) are missing and should

be added. Finally, I would suggest, in order to gain clarity, to summarize section 3.1 in a Table and keep consistency on the use of written numbers.

Reply: we agree summarizing section 3.1 improves the readability of the paper and have moved that information to Table A2 in the appendixes. We apologize for not including the linear regression mentioned in the earlier version but we believe that as this information was not significant, it was therefore with little relevance for the paper and would only add confusion. In this revised version we have therefore decided not to include this analysis as we think we do not have enough data for a good assessment. We have concentrated on time and temperature, and have included this figure in the appendix (Figure A4). The reason to exclude pH is also the fact that due to the metabolic activity of the plants this parameter was highly variable during the measurements (day-night) and the correlation with an average value would not really contribute information on the studied processes.

The results section with the revised statistics and justification for pooling the species-specific data for the sensor part now reads: "Sensor data were collected in the water column, with lateral movement between habitats of water masses, and there were no significant differences, in GPP ($t_{df=31.75}$=-0.16, $p$=0.87), CR ($t_{df=32.46}$=0.91 $p$=0.37) and NCP ($t_{df=32.30}$=0.21, $p$=0.84), between the two species (*P. oceanica* and *C. nodosa*), tested in a mixed model with "Site" as random factor, including depth, region and seasons. Therefore, we didn´t divide the sensor data for the two species."

We have also corrected the abstract and discussion to reflect for instance the fact that, due to using "Site" as random factor in the region analysis, with the high variability, there are no regional differences observed for metabolic rates.

L295-L298. All this info can be removed or moved to the Methods section. If the data is available, please add the correspondent link.

Reply: Information removed and summarized in table A2, in the appendixes. The final database will be available through the repository with the correspondent link upon acceptance of the paper.

L310. In the stats analysis, please provide more details: degrees of freedom, F-values, Sum or Mean of Squares for ANOVA, etc. This information can go in a Table into supplementary materials.

Reply: We have revised the statistics, and updated the results section including t-values (for linear regression models) and $\chi^2$-values (for mixed

models) with accompanying degrees of freedom. We think providing an additional table in the appendix might be confusing as there are many analyses and thus there would have to be several tables or composed tables. We are willing to include these though if the reviewer thinks this would improve the clarity of the paper.

L310. See my general comment above about merging datasets based on simple one-way ANOVAs.

Reply: We agree simple ANOVAs are maybe not the best way to analyse the data. However, we do think that merging some data, is justified. For instance, in the case of the sensor data from the two species. Even though the underlying idea was to capture species-specific metabolic rates, In practice this has proven to be extremely difficult due to lateral movement of water masses. Even in large sandy areas in Posidonia meadows the metabolic signal of the meadow is noticeable (data not shown, personal experience of the authors) and it is difficult to separate the components (species specific productivity) contributing to the ecosystem productivity measured in the water column. So, in this case merging this data has a biologically sound reason, backed up by the statistical test. We did, however, revise the statistics as we do agree the previous analyses were too simple. Nested ANOVAs or ANCOVAs as well as mixed models are far more appropriate. We have decided to use mixed models to be able to include random factors.

L321. See my general comment above about merging datasets based on simple one-way ANOVAs.

Reply: see comments above

L328. Replace "didn´t" by "did not".

Reply: Replaced in the text.

L330-L333. If possible, I would suggest moving the methods and results related to temperature from the appendix to the main manuscript. The finding of temperature not affecting metabolic parameters in the Western basin is very relevant to the work done and is very interesting.

Reply: We agree with the reviewer that the lack of correlation between metabolic parameters for the benthic chamber data and temperature is interesting and definitely unexpected for us. However, we fear this is due to the unbalance of the data over the seasons, with a range of temperatures within different seasons and their corresponding biological activities of the seagrass.

With the revised database we have found relationships with temperature for sensor data, however opposite as expected, with increasing NCP and decreasing CR. Again this could be due to the fact this signal is composed, with many organisms contributing to the measured signal. Also, as for benthic chamber data, the unbalance between seasonal data, with plants in a different growth stage could have influenced the results.

L329. Remove capital letter from "Addition"

Reply: Capital letter removed in the text.

L331. Replace "none" with "any"

Reply: Replaced in the text.

L346. I would suggest removing "and act as carbon sinks" as this was not studied.

Reply: Removed from the text.

L365. Replace "didn´t" by "did not"

Reply: Replaced in the text.

L369. "Except for the summer" hangs alone and it is difficult to know what it means.

Reply: Thank you for the remark, removed in the text.

L373. See my general comment above about merging datasets based on simple one-way ANOVAs.

Reply: as commented above

L375. Keep consistency on the number of decimals used for each parameter.

Reply: Corrected in the text.

**Discussion**

General comment: There are results (I believe from the logistic regressions) written in the Discussion section that should be moved to the Results. Also, it

would help the readers to have a first paragraph on the discussion with the take-home message**.**

Reply: We appreciate your comment. Some of the results presented in the discussion section have been added in the results section. In addition, a first paragraph in the discussion with the take-home message have been added.

L413. Replace "didn´t" by "did not"

Reply: Replaced in the text.

L417. This statement about the 10m distance among seagrass meadows is very confusing. From Table 1, only two sites presented both species. Please clarify what do you mean here.

Reply: Thank you for the remark, clarification added in the text.

L423. Replace "didn´t" by "did not"

Reply: Replaced in the text.

L430. I would suggest removing "and act as carbon sinks".

Reply: Removed in the text.

L432. Keep consistency in the use of acronyms.

Reply: We appreciate your comment and revised all the acronyms in the text.

L439. These results are not presented anywhere.

Reply: Thank you for pointing this out. Now pertinent results have been included in the Results section.

L440 – L447. These results need to be presented in the Results section

Reply: We appreciate your comment. These results have been included in the Results section.

L441. Add space after comma, and remove dot before comma

Reply: Thank you for the remark, corrected in the text.

L446. The results of the biotic parameters related to metabolism are really surprising and it would be interesting to discuss them further.

Reply: As mentioned in the text, biotic parameters like shoot density and biomass were not determinant for GPP, CR nor NCP (p>0.1), which underlines the effect of lateral advection and mixing of water masses influencing the net signal measured by the multiparametric probes. Also, we firmly believe that we lack sufficient data to provide a solid estimate. Therefore we have not included this analysis in the current version of the manuscript. However, we appreciate your comment and believe that this should be included in future studies with more available biotic data.

L454 Replace "wasn´t" by "was not"

Reply: Replaced in the text.

L455. See my comment in methods about the bibliographic research. Does this mean that no benthic chambers have ever been used in C. nodosa in the Eastern basin? If this is the case, the results presented in this work are even more important and this should be highlighted as one of the outcomes.

Reply: We found data on benthic chambers for *Posidonia oceanica* used in the Eastern basin in the publication by Apostolaki et al., 2010, we included the reference below. On the other hand, we did not find published data in the Eastern basin with sensors neither for *P.oceanica* or *C.nodosa*. We have updated the paragraph on the bibliographic search and clarified the method section.

Apostolaki, E. T., Holmer, M., Marbà, N., & Karakassis, I. (2010). Metabolic imbalance in coastal vegetated (*Posidonia oceanica*) and unvegetated benthic ecosystems. *Ecosystems, 13*(3), 459-471.

L458. Replace dot by comma

Reply: Replaced in the text.

L459. Avoid repeating results in the discussion section.

Reply: Removed in the text.

L471. Please cite the correspondent literature.

Reply: Clarified in the text.

L486. Remove dot after column

Reply: Dot removed from the text.

L515. Replace "didn´t" by "did not"

Reply: Replaced in the text.

L518. Remove "a" before "more".

Reply: Removed from the text.

L544. Remove "prevention"

Reply: Removed from the text.

**Appendices**

Appendix B is really scattered and the results of the higher GPP with depth seem to be driven by only 1 depth (15m). Is this only driven by one site?

Reply: Thank you for the remark. We agree there is a high variability in the data. To clarify, the GPP values at 15m depth are measured at the same site for 11 consecutive days and we considered them relatively robust. The

significant relationship of GPP with depth is not present in the new statistical analysis, based on our sanitised database, which is why we have decided to delete the figure.

Appendix D. remove capital letter from oceanica.

Reply: Thank you for the remark, format changed in the text.

**Reply to Comment on bg-2021-60**

**Referee #2**

Comparing the methodology for assessment of GPP using benthic chambers and the potentiometric probes is so apt and need of the hour especially while highlighting the role of marine macrophytes to combat climate change impacts. The aim and objective of the article is genuine and well achieved.

Before detailing our replies to the reviewer I would like to indicate that through the revision of our data for the revised manuscript we have made the hard decision to exclude sensor data with positive oxygen signals during the night time. As we already mentioned in methods, results and discussion sections in the last version, the sensor data has the disadvantage of picking up oxygen concentrations from water volumes drifting past by lateral advection. A positive signal during the night time is a clear indication of this problem and thus we have excluded data where we suspected a big influence of lateral advection. Currents are not usually intense in the Mediterranean and in our opinion this is not a common problem with the dataset and does not invalidate sensor results. However, we have wanted to be on the cautious side and only present data we absolutely confide in. Therefore the database has decreased somewhat in size and this has also meant all statistics and figures have been re-done and some results have changed. We apologize for this, and also the delay it has caused in our revision. The extensive changes made in the manuscript have also caused us to reconsider the order of authors as you may have noticed. However we think the extensive reworking of the manuscript has vastly improved the quality of the analyses and the conclusions are much more robust.

Reply: Thank you very much for your comment. The authors really appreciate it.

Metabolic rates of seagrasses may vary with the temperature, salinity, pH, dissolved oxygen levels etc of ambient water as well as photoperiod and PAR reaching the canopy (depth). If the authors have taken care of these factors prevailed during the long observation period ( 2000 to 2019 while drawing inference, this preprint assumes more merit of publication in the Biogeo Sciences Journal.

Reply: The authors strongly appreciate your comments. The factors mentioned have been taken in account in this study, although not the same amount of associated data was available for all variables and we have not included analyses for which we judged insufficient data was available.

Except for a few typographical errors (page no 2, line 43 name not in italics, Page 1 line 23 Easter or Eastern basin?) and grammer in a few pages (page 11 line 281 tense), the manuscript has been well constructed with bold presentation of results.

Reply: Thank you for the remarks. In order to improve the quality, corrections have been made in the main text.

---

## Referee Report (RR1)

General comments:

The level of changes along the manuscript is large, including a change in authors order, removal of data, change in results and many modifications in the text. Nevertheless, the manuscript is now clearer. I am concerned that the high variability in the data and the low number of samples in some of the analysis could lead to non-robust results (please check the residuals of the statistical analysis). However, the authors are very transparent with this fact and the synthesis effort done is very valuable. Most of my comments are minor, and if achieved, my recommendation is to accept the paper.
* * *
**Abstract:**

- it would really improve by adding the gap in knowledge. I would suggest something like:

Seagrasses can act as carbon sinks, by buffering low pH values during the day and storing carbon in the sediment underneath their meadows. However, available data is scattered and collected using different methodologies making its interpretation and generalization very challenging.

**Introduction:**

Separate paragraphs in L40, L50, L61

L100 – missing closing bracket ")"
L101 – Add comma after increasing
L111 – sometimes oxygen appears as "O2" and other as "oxygen". Keep consistency when possible.

**Methods**

L127 – L129 I suggest the following change in order to increase clarity of the statement:

We compiled data from multiparametric sensors, which was collected during different periods ranging from 2011 to 2019 (for details see Table 1), and data using the benthic chambers methodology, which had a higher number of literature studies with a total of 12 publications for *P. oceanica* and/or *C. nodosa* meadows (for details see Table 2), and a wider temporal cover with studies carried out from 1982 to 2019.

Table 1
- caption: change text to: Characteristics of sampling stations with data for multiparametric sensors. Temperature and salinity represent average values during the deployment.
Table 2
- caption: change text to: Characteristics of sampling stations with data for benthic chamber deployments. Temperature and salinity represent average values during the deployment.
- Add NA to missing values. A lot of sites do not have depth info, is it possible to add this? If exact depth is unknown, is it possible to add a range? For instance, Barron et al. 2004 is (<2 m) and

Barron et al. 2006 is 7m. Does all metabolic data have depth data associated? This is especially relevant as depth is then analyzed as a fixed effect.

L131-132 move sentence "Data available as oxygen concentration over time was processed and analyzed to obtain the metabolic
parameters, when this was not available, we used the reported metabolic rates." to data analysis
L136 remove comma after study
L144 include deployed after were
L147 change Posidonia by *P.oceanica*

L178-L187 I suggest the following change in order to increase clarity in the text:

We compare metabolic data obtained between both methodologies, benthic chambers and multiparametric sensors. For benthic chambers, reported metabolic data in NCP, GPP and CR were extracted from literature as well as accompanying biotic parameters. In these articles metabolic data was generally estimated from changes in dissolved oxygen using the Winkler titration spectrophotometric method (Labasque et al., 2004). Benthic chambers enclose a section of the seagrass meadow, and flexible fitted plastic bags, not permeable for gases, assure the possibility of movement of the seagrass shoots inside, see details in the method section of each paper for the exact construction used. The benthic chamber methodology has been more generally used to assess metabolism of seagrass meadows and the database of this study contains a total of 100 NCP estimations. For multiparametric sensors data, we have calculated metabolism from raw oxygen profiles where possible (including for published and new data described in 2.1), and only used directly reported values for the data obtained from Champenois et al., 2012 and 2019.

L196 add dot after cite.
L277-278 How where the abiotic parameters analyzed? Is a simple linear regression? Are there random factors considered? Add these details.
L279 A common technique for handling negative values is to add a constant value to the data prior to applying the log transform. The transformation is therefore log(Y+a) where a is the constant.
L280 – did you check for normality/homoscedasticity in the residuals of the models?

Remove the part of methods related to pH. Unless I have missed it, pH is not used in any part of the manuscript.

**Results**

Table 1 – this is a great summary table. The word Annual needs correction. I also would suggest removing this row from the table or add annual values for all the cases where data for the four seasons is available. I would suggest including in the caption or in the table that the sensors can provide data for the 2 species together.

L303 – L309 split in two sentences

L323 It is unclear if the following sentence "tested in a mixed model with "Site" as random factor, including depth, region and seasons" means that the model used 3 fixed effects (2 categorical and 1

continuous "depth") in the form Y = Species + region + season + depth + (1|site). This could be described in the methods or by adding this info after seasons.

L324 change didn't to did not.

Caption of Figure 5 – include that there were no differences

**Discussion**

L412 change was to were
L413 add an S to quarter
L422 remove comma before masking
L438 for some reason for is strikethrough.
L456 -458 any potential explanation for this result? It is really surprising.
L510 use independently
L516 -520 Could you add cites for the lack of historical marked sites and temperature effect in *C.nodosa*?

**Appendix and Supplementary**

I cannot access these materials so I cannot evaluate them. What is confusing to me is if there are two supplementary files (an appendix and a supplement) or just one. I ask this because sometimes figures are referred as Fig 5A and others Fig1S

---

## Referee Report (RR2)

**Affiliation supercript 2** is repeated

**Methods**

Include, in the section for stats, the analysis over time

L340 specify that for NCP the residuals of the models achieved a normal distribution, although not transformed.

**Results**

All the stats for lmer are reported as contrasts of the fixed part ($t_{df=\ldots} = \ldots$, $p = \ldots$) which is the outcome of the summary() function in the lme4 package in R. However, this does not show the significance of the fixed effect being assessed in the model, just if there are specific differences between the intercepts and the factor categories (for categorical variables such as Region, Species, Season). In  most of the analysis in this manuscript the fixed factors only have 2 categories, and so, this is likely not a problem as the stats reported are already showing differences between the two categories. However, in the case of Season there are 4 categories, and when the authors report "NCP was lower in spring ($t_{df=23.89} = -3.69$, $p < 0.01$)" they are not reporting that there are differences among seasons. What they report is that spring is different from the intercept (I assume the intercept will be determined by Fall based on alphabetical order, although I am not sure what order the authors used in the code) and that the other categories are not different from the intercept.

I would recommend including the results table for the fixed factors, which can be obtained by using anova() function in R. For instance, in a model like "m1=lmer(metabolic rate ~ Region + Depth + Season + (1|Site))" use anova (m1) to report:

Region (DF=1, sum squares, mean squares, F-value, p=…)

Depth (as continuous) (DF=1, , sum squares, mean squares, F-value, p=…)

Season (DF=3, sum squares, mean squares, F-value, p=…)

And then use summary(m1) to report which specific categories are showing these differences, as it is already done.

I hope the following picture helps:

```
> sensors=subset(data,data$Methodology=="sensors")
> s=lmer(NCP~Region+Depth+Season+(1|Site),data=sensors)
> anova(s)
Type III Analysis of Variance Table with Satterthwaite's method
       Sum Sq Mean Sq NumDF  DenDF F value     Pr(>F)
Region  71404   71404     1  3.925 11.3725 0.0288098 *
Depth    3712    3712     1 10.293  0.5912 0.4592277
Season 197898   65966     3 25.649 10.5064 0.0001091 ***
* * *
Signif. codes:  0 '***' 0.001 '**' 0.01 '*' 0.05 '.' 0.1 ' ' 1
> summary(s)
Linear mixed model fit by REML. t-tests use Satterthwaite's method [lmerModLmerTest]
Formula: NCP ~ Region + Depth + Season + (1 | Site)
   Data: sensors

REML criterion at convergence: 415.6

Scaled residuals:
    Min      1Q  Median      3Q     Max
-3.5728 -0.4169  0.0610  0.5374  1.6511

Random effects:
 Groups   Name        Variance Std.Dev.
 Site     (Intercept) 10704    103.46
 Residual              6279     79.24
Number of obs: 40, groups:  Site, 7

Fixed effects:
             Estimate Std. Error      df t value Pr(>|t|)
(Intercept)  -244.314    125.738   5.585  -1.943  0.10360
RegionWEST    408.704    121.194   3.925   3.372  0.02881 *
Depth           6.531      8.493  10.293   0.769  0.45923
SeasonSpring -302.430     82.321  20.276  -3.674  0.00148 **
SeasonSummer  -41.255     68.630  20.695  -0.601  0.55428
SeasonWinter  -21.470     89.843  27.650  -0.239  0.81289
* * *
Signif. codes:  0 '***' 0.001 '**' 0.01 '*' 0.05 '.' 0.1 ' ' 1

Correlation of Fixed Effects:
            (Intr) RgWEST Depth  SsnSpr SsnSmm
RegionWEST  -0.829
Depth       -0.183 -0.092
SeasonSprng -0.153  0.082 -0.569
SeasonSummr -0.238  0.129 -0.602  0.814
SeasonWintr -0.128  0.087 -0.536  0.651  0.736
>
```

I am positive, based on the plots and the data in the tables that the results will be similar to what is already written, but the missing information is essential for readers to follow the process that the authors went through. If it is easier for the authors, they can add the results table from anova() and summary() in supplementary.

**Other minor issues:**

**Table 2:**

Missing depth in Alcanada ID 21.

Explain in the caption the difference between Yearly and Av. Year.

**Table 3:** There is an  empty cell in the first column of benthic chambers - *Cymodocea nodosa* that I suspect corresponds to "Winter". Please add the missing label.

**Figure 4:** write the species names in italics

L235 pH sampling information can be removed

L369 replace statement "with as only factor methodology and as random effect study" with "with methodology as fixed factor and publication as random"

Fig S2a and S2b: Annual is missing an "n"

L380 typo in the p-value

L547 typo in "*Cymodcea nodosa*". Actually, the species name can be abbreviated

*L560 R or CR?*

L581. There is (again) a reference to an inexistent appendix. Please check the text carefully to avoid these types of mistakes.

L624 remove capital letters in Eddy Covariance

L705 check that all species names are in italics

Along the text sometimes the authors use the term "study" and others "publication" to describe on of the random factors. I would suggest keeping consistency in the terms.

Along the text there is inconsistent use of acronyms. For instance, in *L576 "*Gross Productivity and Community Respiration" are used while GPP and CR have been already used before.

---

## Author Response (AR2)

**Abstract**

Comment:
- it would really improve by adding the gap in knowledge. I would suggest something like:

Seagrasses can act as carbon sinks, by buffering low pH values during the day and storing carbon in the sediment underneath their meadows. However, available data is scattered and collected using different methodologies making its interpretation and generalization very challenging.

> Reply:
> We have added a slightly modified version of the suggested sentence as: "Seagrasses can act as carbon sinks; buffer lowering pH values during the day and storing carbon in the sediment underneath their meadows. However, available data documenting these processes is scattered and collected using different methodologies, which makes its interpretation and generalization very challenging."

**Introduction**

Comments:
Separate paragraphs in L40, L50, L61

> Reply: We have introduced separations at the line numbers of the original document (now L41, L52, L64)

L100 – missing closing bracket "("
L101 – Add comma after increasing
L111 – sometimes oxygen appears as "O2" and other as "oxygen". Keep consistency when possible.

> Reply: L100 (now L105), we have deleted the opening bracket in L103 to improve readability of the sentence. We added the comma after increasing in L105 (before L101). We have changed all mention of the formula ($O_2$) to "oxygen" in the text unless specific values with units (i.e. mmol $O_2$ $day^{-1}$) are mentioned.

**Methods**

Comments:
L127 – L129 I suggest the following change in order to increase clarity of the statement:

We compiled data from multiparametric sensors, which was collected during different periods ranging from 2011 to 2019 (for details see Table 1), and data using the benthic chambers methodology, which had a higher number of literature studies with a total of 12 publications for *P. oceanica* and/or *C. nodosa* meadows (for details see Table 2), and a wider temporal cover with studies carried out from 1982 to 2019.

> Reply: we have changes the sentence as suggested, now L134-137 in the revised document.

Table 1
- caption: change text to: Characteristics of sampling stations with data for multiparametric sensors. Temperature and salinity represent average values during the deployment.

       Reply: changed as suggested.

Table 2
- caption: change text to: Characteristics of sampling stations with data for benthic chamber deployments. Temperature and salinity represent average values during the deployment.

       Reply: changed as suggested.

- Add NA to missing values. A lot of sites do not have depth info, is it possible to add this? If exact depth is unknown, is it possible to add a range? For instance, Barron et al. 2004 is (<2 m) and Barron et al. 2006 is 7m. Does all metabolic data have depth data associated? This is especially relevant as depth is then analyzed as a fixed effect.

       Reply: We agree with the reviewer that depth is crucial information. We have gone back to each individual article and extracted the information about depth that somehow had got lost during compilation and processing of the database. We have used discrete values, i.e. for Barron et al. 2004 we have used 2m depth, even though the article states <2m, as this allows for the use of depth as a continuous variable in a regression analysis. We have re-done the statistic model using all data available for depth, therefore some changes in reported statistical values have occurred but nothing changing the results.

L131-132 move sentence "Data available as oxygen concentration over time was processed and analysed to obtain the metabolic parameters, when this was not available, we used the reported metabolic rates."

       Reply: we have moved this sentence to data analysis and integrated it there combining with a similar sentence containing the same information (L210-214).

L136 remove comma after study
L144 include deployed after were
L147 change Posidonia by *P.oceanica*

       Reply: revised the text as suggested

L178-L187 I suggest the following change in order to increase clarity in the text:
We compare metabolic data obtained between both methodologies, benthic chambers and multiparametric sensors. For benthic chambers, reported metabolic data in NCP, GPP and CR were extracted from literature as well as accompanying biotic parameters. In these articles metabolic data was generally estimated from changes in dissolved oxygen using the Winkler titration spectrophotometric method (Labasque et al., 2004). Benthic chambers enclose a section of the seagrass meadow, and flexible fitted plastic bags, not permeable for gases, assure the possibility of movement of the seagrass shoots inside, see details in the method section of each paper for the exact construction used. The benthic chamber methodology has been more generally used to assess metabolism of seagrass meadows and the database of this study contains a total of 100 NCP estimations. For multiparametric sensors data, we have calculated metabolism from raw oxygen

profiles where possible (including for published and new data described in 2.1), and only used directly reported values for the data obtained from Champenois et al., 2012 and 2019.

> Reply: We thank the reviewer for the suggestion, which we have largely followed. The text now reads: "We compare metabolic data obtained by both methods, benthic chambers and multiparametric sensors. For benthic chambers, reported metabolic data as well as accompanying biotic parameters were extracted from the literature.  In these articles Net Community Production (NCP) and Respiration (R) was generally estimated from changes in dissolved oxygen using the Winkler titration spectrophotometric method (Labasque et al., 2004). Benthic chambers enclose a section of the seagrass meadow, and flexible fitted plastic bags, not permeable for gases, assure the possibility of movement of the shoots inside, see details in the method section of each paper for the exact construction used.  The benthic chamber methodology has been more generally used to assess metabolism of seagrass meadows and the database of this study contains a total of 100 NCP estimations. For multiparametric sensor data, data available as oxygen concentration over time was processed and analysed to obtain the metabolic parameters (see section 2.3.1), when this was not available, we used the reported metabolic rates. We compare the data obtained between both methodologies, with calculated metabolism from raw oxygen profiles obtained with the multiparametric sensors where possible, and only used directly reported productivity values for the sensor data obtained from Champenois et al., 2012 and 2019."

L196 add dot after cite.
L277-278 How where the abiotic parameters analyzed? Is a simple linear regression? Are there random factors considered? Add these details.
L279 A common technique for handling negative values is to add a constant value to the data prior to applying the log transform. The transformation is therefore log(Y+a) where a is the constant.
L280 – did you check for normality/homoscedasticity in the residuals of the models?
Remove the part of methods related to pH. Unless I have missed it, pH is not used in any part of the manuscript.

> Reply: L277-278: yes that is correct, we fitted individual regression models for the abiotic parameters with site as a rando factor. We have added the model formula to the text for clarification.
> L279: Yes, we have tried transforming NCP by adding 100 (log(NCP+100)) to avoid negative values. However, this did not give a better result compared to using non-transformed values, as data were not normal distributed either. As a normal distribution is not a strict requisite we checked the residuals of each model by plotting them and assured a normal distribution for all models, also non-transformed NCP.
> We removed the methodology related to pH.  we did not pursue the pH as a factor in our analysis as this is too dependent on the measurement time. The mention of this parameter was a remnant of a previous version of the manuscript, we have now removed this.

**Results**

Comments:
Table 1 – this is a great summary table. The word Annual needs correction. I also would suggest removing this row from the table or add annual values for all the cases where data for the four seasons is available. I would suggest including in the caption or in the table that the sensors can provide data for the 2 species together.

Reply: We have replaced the word "annual", we have not removed it from the table as some studies do not distinguish between seasons and just give a yearly average. We had included both species within the same limitation (between the same lines) in Table 1 with sensor data to reflect the measurement site and specific habitat but have not included the observation that sensors do not pick-up the species specific signal in the legend as we included this information in the results.

L303 – L309 split in two sentences

Reply: We have split the sentence in three for clarity. The paragraph now reads: "Benthic chambers and multiparametric sensors yielded very different CR with 41.2 ± 4.55 (SE) mmol $O_2$ m$^{-2}$ d$^{-1}$ for benthic chambers and 229.9 ± 25.57 mmol $O_2$ m$^{-2}$ d$^{-1}$ for sensors ($t_{df=84.86}$=9.57, $p$<0.0001) in a mixed model, with as only factor methodology and as random effect study. This difference with almost an order of magnitude is found for NCP as well with 18.8 ± 3.80 and 143.2 ± 28.21 mmol $O_2$ m$^{-2}$ d$^{-1}$ for benthic chambers and sensors respectively ($t_{df=25.61}$=2.78, $p$<0.001) as well as for GPP (55.3 ± 6.39 and 329.2 ± 29.91 mmol $O_2$ m$^{-2}$ d$^{-1}$ for chambers and sensors; $t_{df=101.05}$=11.14, $p$<0.0001) (Fig. 2). Therefore we decided to analyse the metabolic rates estimated using benthic chambers and multiparametric sensors separately."

L323 It is unclear if the following sentence "tested in a mixed model with "Site" _as random factor, including depth, region and seasons" _means that the model used 3 fixed effects (2 categorical and 1 continuous "depth") in the form Y = Species + region + season + depth + (1|site). This could be described in the methods or by adding this info after seasons.

Reply: We have added the model formulation to the text to clarify what model was used.

L324 change didn't to did not.

Reply: changed

Caption of Figure 5 – include that there were no differences

Reply: added

**Discussion**

Comments:
L412 change was to were
L413 add an S to quarter
L422 remove comma before masking
L438 for some reason for is strikethrough.
L456 -458 any potential explanation for this result? It is really surprising.
L510 use independently
L516 -520 Could you add cites for the lack of historical marked sites and temperature effect in *C.nodosa*?

Reply: L412, L413, L422, L438 changed as suggested.
L456-458: We think differences and local adaptation to temperature between sites might have been bigger than a clear relationship of CR and temperature. We have added a sentence: "There is no data available for intermediate temperatures, leaving two clusters,

one between 13 and 16 °C and one between 23 and 27°C. The bulk of the data is collected between 23 and 27 °C, therefore dominating the regression while at lower temperatures a clear increase in CR is visible between 13 and 16 °C. Differences between sites for summer measurements might obscure a possible relationship of CR and temperature."

**Appendix and Supplementary**

I cannot access these materials so I cannot evaluate them. What is confusing to me is if there are two supplementary files (an appendix and a supplement) or just one. I ask this because sometimes figures are referred as Fig 5A and others Fig1S

> Reply: there is only 1 supplementary file, we might have introduced confusing references to this file. We have tried to clarify the text and references, referring to Fig. XS now instead of Fig. XA. The full database on which the paper is based is available in a repository. We have now deposited the file and it should be shortly publicly accessible at https://digital.csic.es/. Due to problems with the online submission system of that repository it is submitted by email, and could take a few days to be online. We will include the handle in the final article.

---

## Author Response (AR3)

*Replies indented in blue*

**Affiliation supercript 2** is repeated

*Reply: Correct, the third affiliation should be 3 (Marilles), changed. Thank you for noticing*

**Methods**
Include, in the section for stats, the analysis over time

*Reply: we added a sentence to the end of the statistical analysis section "Finally, we evaluated if trends in metabolic rates over time were apparent with simple lineal regressions."*

L340 specify that

*Reply: "that" was part of the sentence structure "confirming that", with the second part of the sentence the confirmation. We have clarified the sentence, it now reads: "confirming the general assumption that seagrass meadows normally tend to be autotrophic ecosystems with a mean P/R ratio above 1"*

**Results**
All the stats for lmer are reported as contrasts of the fixed part ($t_{df=\ldots}=\ldots$, $p=\ldots$) which is the outcome of the summary() function in the lme4 package in R. However, this does not show the significance of the fixed effect being assessed in the model, just if there are specific differences between the intercepts and the factor categories (for categorical variables such as Region, Species, Season). In most of the analysis in this manuscript the fixed factors only have 2 categories, and so, this is likely not a problem as the stats reported are already showing differences between the two categories. However, in the case of Season there are 4 categories, and when the authors report "NCP was lower in spring ($t_{df=23.89}=-3.69$, $p<0.01$)" they are not reporting that there are differences among seasons. What they report is that spring is different from the intercept (I assume the intercept will be determined by Fall based on alphabetical order, although I am not sure what order the authors used in the code) and that the other categories are not different from the intercept.

I would recommend including the results table for the fixed factors, which can be obtained by using anova() function in R. For instance, in a model like "m1=lmer(metabolic rate ~ Region + Depth + Season + (1|Site))" use anova (m1) to report:

Region (DF=1, sum squares, mean squares, F-value, $p=\ldots$)

Depth (as continuous) (DF=1, , sum squares, mean squares, F-value, $p=\ldots$)

Season (DF=3, sum squares, mean squares, F-value, $p=\ldots$)

And then use summary(m1) to report which specific categories are showing these differences, as it is already done.

I hope the following picture helps:

```
> sensors=subset(data,data$Methodology=="sensors")
> s=lmer(NCP~Region+Depth+Season+(1|Site),data=sensors)
> anova(s)
Type III Analysis of Variance Table with Satterthwaite's method
        Sum Sq Mean Sq NumDF  DenDF F value     Pr(>F)
Region   71404   71404     1  3.925 11.3725 0.0288098 *
Depth     3712    3712     1 10.293  0.5912 0.4592277
Season  197898   65966     3 25.649 10.5064 0.0001091 ***
* * *
Signif. codes:  0 '***' 0.001 '**' 0.01 '*' 0.05 '.' 0.1 ' ' 1
> summary(s)
Linear mixed model fit by REML. t-tests use Satterthwaite's method [lmerModLmerTest]
Formula: NCP ~ Region + Depth + Season + (1 | Site)
   Data: sensors

REML criterion at convergence: 415.6

Scaled residuals:
    Min      1Q  Median      3Q     Max
-3.5728 -0.4169  0.0610  0.5374  1.6511

Random effects:
 Groups   Name        Variance Std.Dev.
 Site     (Intercept) 10704    103.46
 Residual              6279     79.24
Number of obs: 40, groups:  Site, 7

Fixed effects:
             Estimate Std. Error       df t value Pr(>|t|)
(Intercept)  -244.314    125.738    5.585  -1.943  0.10360
RegionWEST    408.704    121.194    3.925   3.372  0.02881 *
Depth           6.531      8.493   10.293   0.769  0.45923
SeasonSpring -302.430     82.321   20.276  -3.674  0.00148 **
SeasonSummer  -41.255     68.630   20.695  -0.601  0.55428
SeasonWinter  -21.470     89.843   27.650  -0.239  0.81289
* * *
Signif. codes:  0 '***' 0.001 '**' 0.01 '*' 0.05 '.' 0.1 ' ' 1

Correlation of Fixed Effects:
            (Intr) RgWEST Depth  SsnSpr SsnSmm
RegionWEST  -0.829
Depth       -0.183 -0.092
SeasonSprng -0.153  0.082 -0.569
SeasonSummr -0.238  0.129 -0.602  0.814
SeasonWintr -0.128  0.087 -0.536  0.651  0.736
> |
```

I am positive, based on the plots and the data in the tables that the results will be similar to what is already written, but the missing information is essential for readers to follow the process that the authors went through. If it is easier for the authors, they can add the results table from anova() and summary() in supplementary.

> Reply: We thank the reviewer for this suggestion, which we have incorporated. Adding an ANOVA did not change the statistic outcome, for instance for the first test looking at differences between methods, that had only 1 fixed effect. In fact, where we reported $\chi 2$ values before, we had compared our lmer model with a model where the fixed effect was set to the value 1 and then we ran an ANOVA on both models to obtain the significance. This was not clearly explained in the methods and the suggested solution is far more practical.
>
> We have revised the method and the results sections accordingly and produced a summary table in the supplementary material for statistics.

**Other minor issues:**

**Table 2:**
Missing depth in Alcanada ID 21.

> Reply: we thank the reviewer for noticing this detail, we have added the missing value.

Explain in the caption the difference between Yearly and Av. Year

Reply: Yearly represents average values for different years. We agree with the reviewer this is confusing as there is only 1 dataset with repetitive measurements and the range is given in the column "year". We have changed "Yearly" to "Av. Year" for consistency.

**Table 3:** There is an empty cell in the first column of benthic chambers - *Cymodocea nodosa* that I suspect corresponds to "Winter". Please add the missing label.

Reply: actually, we did not find data for *C. nodosa* for winter and the line behind the species name is the average of all measured values for *C. nodosa*. As we only found values for the Western Mediterranean these overall averages are the same as the average for the Western Mediterranean and the first line is equal to the second line where that is summarized. We have clarified the Table by removing this obsolete line of data but left the repeated information for the sensors in the East, where we only had one season (summer) and the overall values repeat the seasonal values.

**Figure 4:** write the species names in italics

Reply: we have successfully changed the code for the graphs and species names are now in italics in the figures.

L235 pH sampling information can be removed

Reply: removed

L369 replace statement "with as only factor methodology and as random effect study" with "with methodology as fixed factor and publication as random"

Reply: sentence changed according to indications

Fig S2a and S2b: Annual is missing an "n"

Reply: Thank you for noticing. Also Table S2 contained the same error. We have changed "Annual" to "Annual" in all appendix material.

L380 typo in the p-value

Reply: we have used p<0.01 as well as p<0.001 to indicate different confident intervals with as limit for significance p<0.05.

L547 typo in "*Cymodcea nodosa*". Actually, the species name can be abbreviated

Reply: correct, abbreviated.

*L560 R or CR?*

Reply: CR as defined in line 76 (of the last submitted version): "Community Respiration (CR)." We have changed the mistake (R to CR).

L581. There is (again) a reference to an inexistent appendix. Please check the text carefully to avoid these types of mistakes.

Reply: This figure in the appendix has changed in numbering from A6 to S3. It is an existing Figure, however the reference to the appendix was mistaken. We have changed this reference and revised the text

L624 remove capital letters in Eddy Covariance

Reply: changed to normal font

L705 check that all species names are in italics

Reply: changed in the reference of Gacia et al., 2012, and revised the rest of the list.

Along the text sometimes the authors use the term "study" and others "publication" to describe on of the random factors. I would suggest keeping consistency in the terms

Reply: changed in two occasions to clarify and unify the text.

Along the text there is inconsistent use of acronyms. For instance, in *L576* "Gross Productivity and Community Respiration" are used while GPP and CR have been already used before.

Reply: We had repeated the full wording to remind the reader of the terminology in the discussion section. As we indeed had specified the abbreviation before we have now included the abbreviations as well at first mention of the full wording in the discussion so it is absolutely clear we are referring to the same concept.